# LeMoNADe: Learned Motif and Neuronal Assembly Detection in calcium imaging videos

**Elke Kirschbaum**[1]     **Manuel Haußmann**[1]     **Steffen Wolf**[1]
{elke.kirschbaum,manuel.haussmann,steffen.wolf}@iwr.uni-heidelberg.de

**Hannah Sonntag**[2]
hannah.sonntag@mpimf-heidelberg.mpg.de

**Justus Schneider**[3]     **Shehabeldin Elzoheiry**[3]
{justus.schneider, shehab.elzoheiry}@physiologie.uni-heidelberg.de

**Oliver Kann**[3]
oliver.kann@physiologie.uni-heidelberg.de

**Daniel Durstewitz**[4]                        **Fred A. Hamprecht**[1]
daniel.durstewitz@zi-mannheim.de          fred.hamprecht@iwr.uni-heidelberg.de

[1]Interdisciplinary Center for Scientific Computing (IWR), Heidelberg University, Germany
[2]Institute for Anatomy and Cell Biology, Heidelberg University, Germany
[3]Institute of Physiology and Pathophysiology, Heidelberg University, Germany
[4]Dept. Theoretical Neuroscience, Central Institute of Mental Health, Mannheim, Germany

## Abstract

Neuronal assemblies, loosely defined as subsets of neurons with reoccurring spatio-temporally coordinated activation patterns, or "motifs", are thought to be building blocks of neural representations and information processing. We here propose LeMoNADe, a new exploratory data analysis method that facilitates hunting for motifs in calcium imaging videos, the dominant microscopic functional imaging modality in neurophysiology. Our nonparametric method extracts motifs directly from videos, bypassing the difficult intermediate step of spike extraction. Our technique augments variational autoencoders with a discrete stochastic node, and we show in detail how a differentiable reparametrization and relaxation can be used. An evaluation on simulated data, with available ground truth, reveals excellent quantitative performance. In real video data acquired from brain slices, with no ground truth available, LeMoNADe uncovers nontrivial candidate motifs that can help generate hypotheses for more focused biological investigations.

## 1 Introduction

Seventy years after being postulated by Hebb (1949), the existence and importance of reoccurring spatio-temporally coordinated neuronal activation patterns (motifs), also known as neuronal assemblies, is still fiercely debated (Marr et al., 1991; Singer, 1993; Nicolelis et al., 1997; Ikegaya et al., 2004; Cossart & Sansonetti, 2004; Buzsáki, 2004; Mokeichev et al., 2007; Pastalkova et al., 2008; Stevenson & Kording, 2011; Ahrens et al., 2013; Carrillo-Reid et al., 2015). Calcium imaging, a microscopic video technique that enables the concurrent observation of hundreds of neurons in vitro and in vivo (Denk et al., 1990; Helmchen & Denk, 2005; Flusberg et al., 2008), is best suited to witness such motifs if they indeed exist.

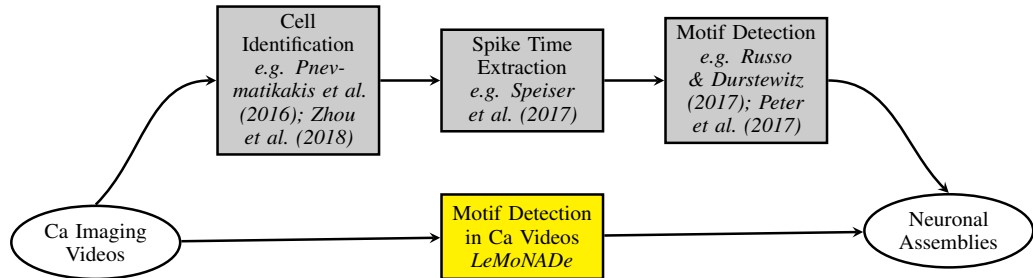

Figure 1: We present *LeMoNADe*, a novel approach to identify neuronal assemblies directly from calcium imaging data. In contrast to previous methods, LeMoNADe does not need pre-processing steps such as cell identification and spike time extraction for unravelling assemblies.

In recent years, a variety of methods have been developed to identify neuronal assemblies. These methods range from approaches for the detection of synchronous spiking, up to more advanced methods for the detection of arbitrary spatio-temporal firing patterns (Comon, 1994; Nicolelis et al., 1995; Grün et al., 2002a;b; Lopes-dos Santos et al., 2013; Russo & Durstewitz, 2017; Peter et al., 2017). All of these methods, however, require a spike time matrix as input. Generating such a spike time matrix from calcium imaging data requires the extraction of individual cells and discrete spike times. Again, many methods have been proposed for these tasks (Mukamel et al., 2009; Pnevmatikakis & Paninski, 2013; Pnevmatikakis et al., 2013; Diego et al., 2013; Diego & Hamprecht, 2013; Pachitariu et al., 2013; Pnevmatikakis et al., 2014; Diego & Hamprecht, 2014; Kaifosh et al., 2014; Pnevmatikakis et al., 2016; Apthorpe et al., 2016; Inan et al., 2017; Spaen et al., 2017; Klibisz et al., 2017; Speiser et al., 2017; Zhou et al., 2018). Given the low signal-to-noise ratios (SNR), large background fluctuations, non-linearities, and strong temporal smoothing due to the calcium dynamics itself as well as that of calcium indicators, it is impressive how well some of these methods perform, thanks to modern recording technologies and state-of-the-art regularization and inference (Pnevmatikakis et al., 2016; Zhou et al., 2018). Still, given the difficulty of this data, errors in segmentation and spike extraction are unavoidable, and adversely affect downstream processing steps that do not have access to the raw data. Hence, properly annotating data and correcting the output from automatic segmentation can still take up a huge amount of time.

In this paper, we propose *LeMoNADe (Learned Motif and Neuronal Assembly Detection)*, a variational autoencoder (VAE) based framework specifically designed to identify repeating firing motifs with arbitrary temporal structure directly in calcium imaging data (see figure 1). The encoding and decoding networks are set up such that motifs can be extracted directly from the decoding filters, and their activation times from the latent space (see sec. 3). Motivated by the sparse nature of neuronal activity we replace the Gaussian priors used in standard VAE. Instead we place Bernoulli priors on the latent variables to yield sparse and sharply peaked motif activations (sec. 3.1). The choice of discrete Bernoulli distributions makes it necessary to use a BinConcrete relaxation and the Gumbel-softmax reparametrization trick (Maddison et al., 2016; Jang et al., 2017) to enable gradient descent techniques with low variance (sec. 3.3). We add a $\beta$-coefficient (Higgins et al., 2017) to the loss function in order to adapt the regularization to the properties of the data (sec. 3.3). Furthermore, we propose a training scheme which allows us to process videos of arbitrary length in a computationally efficient way (sec. 3.4). On synthetically generated datasets the proposed method performs as well as a state-of-the-art motif detection method that requires the extraction of individual cells (sec. 4.1). Finally, we detect possible repeating motifs in two fluorescent microscopy datasets from hippocampal slice cultures (sec. 4.2). A PyTorch implementation of the proposed method is released at `https://github.com/EKirschbaum/LeMoNADe`.

## 2    RELATED WORK

**Autoencoder and variational autoencoder**    Variational Autoencoders (VAEs) were introduced by Kingma & Welling (2014) and have become a popular method for unsupervised generative deep learning. They consist of an encoder, mapping a data point into a latent representation, and a decoder whose task is to restore the original data and to generate samples from this latent space. However, the

original VAE lacks an interpretable latent space. Recent suggestions on solving this problem have been modifications of the loss term (Higgins et al., 2017) or a more structured latent space (Johnson et al., 2016; Deng et al., 2017).

VAE have also been successfully used on video sequences. Li & Mandt (2018) learn a disentangled representation to manipulate content in cartoon video clips, while Goyal et al. (2017) combine VAEs with nested Chinese Restaurant Processes to learn a hierarchical representation of video data. Johnson et al. (2016) use a latent switching linear dynamical system (SLDS) model combined with a structured variational autoencoder to segment and categorize mouse behavior from raw depth videos. Unfortunately, this model is not directly applicable to the task of identifying motifs with temporal structure from calcium imaging data for the following reasons: Firstly, neuronal assemblies are expected to extend over multiple frames. Since in the model by Johnson et al. (2016) the underlying latent process is a relatively simple first-order Markovian (switching) linear process, representing longer-term temporal dependencies will be very hard to achieve due to the usually exponential forgetting in such systems. Secondly, in the model of Johnson et al. (2016) each frame is generated from exactly one of $M$ latent states. For calcium imaging, however, most frames are not generated by one of the $M$ motifs but from noise, and different motifs could also temporally overlap which is also not possible in the model by Johnson et al. (2016).

Closest to our goal of detecting motifs in video data is the work described in Bascol et al. (2016). In this approach, a convolutional autoencoder is combined with a number of functions and regularization terms to enforce interpretability both in the convolutional filters and the latent space. This method was successfully used to detect patterns in data with document structure, including optical flow features of videos. However, as the cells observed in calcium imaging are spatially stationary and have varying luminosity, the extraction of optical flow features makes no sense. Hence this method is not applicable to the task of detecting neuronal assemblies in calcium imaging data.

**Cell segmentation and spike time extraction from calcium imaging data**     Various methods have been proposed for automated segmentation and signal extraction from calcium imaging data. Most of them are based on non-negative matrix factorization (Mukamel et al., 2009; Pnevmatikakis & Paninski, 2013; Pnevmatikakis et al., 2013; 2014; Diego & Hamprecht, 2014; Pnevmatikakis et al., 2016; Inan et al., 2017; Zhou et al., 2018), clustering (Kaifosh et al., 2014; Spaen et al., 2017), and dictionary learning (Diego et al., 2013; Diego & Hamprecht, 2013; Pachitariu et al., 2013). Recent approaches started to use deep learning for the analysis of calcium imaging data. Apthorpe et al. (2016) and Klibisz et al. (2017) use convolutional neural networks (CNNs) to identify neuron locations and Speiser et al. (2017) use a VAE combined with different models for calcium dynamics to extract spike times from the calcium transients.

Although many sophisticated methods have been proposed, the extraction of cells and spike times from calcium imaging data can still be prohibitively laborious and require manual annotation and correction, with the accuracy of these methods being limited by the quality of the calcium recordings. Furthermore, some of the mentioned methods are specially designed for two-photon microscopy, whereas only few methods are capable to deal with the low SNR and large background fluctuations in single-photon and microendoscopic imaging (Flusberg et al., 2008; Ghosh et al., 2011). Additional challenges for these methods are factors such as non-Gaussian noise, non-cell background activity and seemingly overlapping cells which are out of focus (Inan et al., 2017).

**Neuronal assembly detection**     The identification of neuronal assemblies in spike time matrices has been studied from different perspectives. For the detection of joint (strictly synchronous) spike events across multiple neurons, rather simple methods based on PCA or ICA have been proposed (Comon, 1994; Nicolelis et al., 1995; Lopes-dos Santos et al., 2013), as well as more sophisticated statistical methods such as unitary event analysis (Grün et al., 2002a;b). Higher-order correlations among neurons and sequential spiking motifs such as synfire chains can be identified using more advanced statistical tests (Staude et al., 2010a;b; Gerstein et al., 2012). The identification of cell assemblies with arbitrary spatio-temporal structure has been addressed only quite recently. One approach recursively merges sets of units into larger groups based on their joint spike count probabilities evaluated across multiple different time lags (Russo & Durstewitz, 2017). Another method uses sparse convolutional coding (SCC) for reconstructing the spike matrix as a convolution of spatio-temporal motifs and their activations in time (Peter et al., 2017). An extension of this method uses a group sparsity regularization to identify the correct number of motifs (Mackevicius et al., 2018).

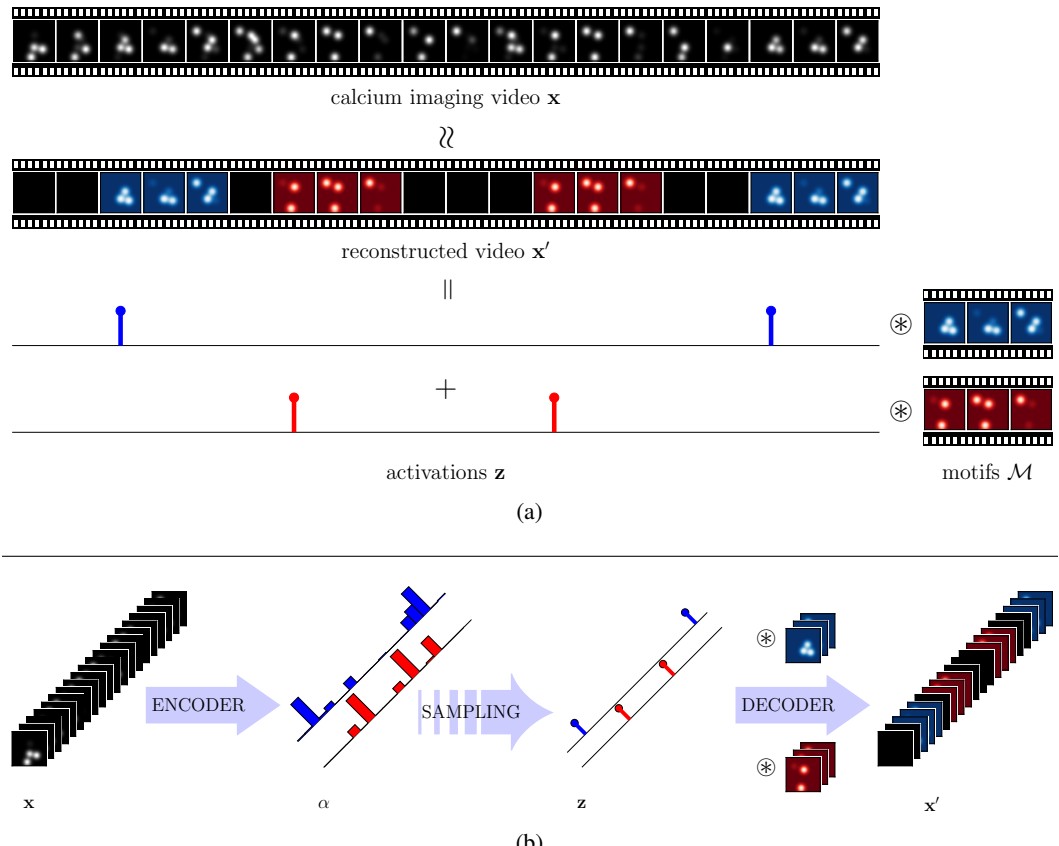

Figure 2: Schematic sketch of the proposed method. In this toy example, the input video $\mathbf{x}$ is an additive mixture of two motifs (highlighted in red and blue) plus noise, as shown in (a). To learn the motifs and activations, the loss between input video $\mathbf{x}$ and reconstructed video $\mathbf{x}'$ is minimized. (b) shows the generation of the reconstructed video through the proposed VAE framework.

To the authors' knowledge, solely Diego & Hamprecht (2013) address the detection of neuronal assemblies directly from calcium imaging data. This method, however, only aims at identifying synchronously firing neurons, whereas the method proposed in this paper can identify also assemblies with more complex temporal firing patterns.

## 3  METHOD

LeMoNADe is a VAE based latent variable method, specifically designed for the unsupervised detection of repeating motifs with temporal structure in video data. The data $\mathbf{x}$ is reconstructed as a convolution of motifs and their activation time points as displayed in figure 2a. The VAE is set up such that the latent variables $\mathbf{z}$ contain the activations of the motifs, while the decoder encapsulates the firing motifs of the cells as indicated in figure 2b. The proposed generative model is displayed in figure 3. The great benefit of this generative model in combination with the proposed VAE is the possibility to directly extract the temporal motifs and their activations and at the same time take into account the sparse nature of neuronal assemblies.

### 3.1  THE LeMoNADe MODEL

In the proposed model the dataset consists of a single video $\mathbf{x} \in \mathbb{R}^{T \times P \times P'}$ with $T$ frames of $P \times P'$ pixels each. We assume this video to be an additive mixture of $M$ repeating motifs of maximum temporal length $F$. At each time frame $t = 1, \ldots, T$, and for each motif $m = 1, \ldots, M$, a latent random variable $z_t^m \in \{0, 1\}$ is drawn from a prior distribution $p_a(\mathbf{z})$. The variable $z_t^m$ indicates

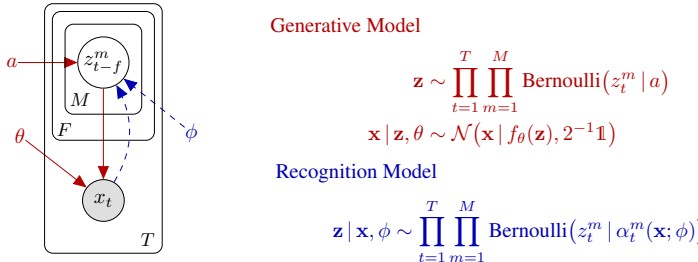

Figure 3: Plate diagram and proposed generative and recognition model. We show the plate diagram of the proposed model (left), where red (solid) lines correspond to the generative/decoding process and blue (dashed) lines correspond to the recognition/encoding model. On the right the equations for the generative as well as the recognition model are given.

whether motif $m$ is activated in frame $t$ or not. The video $\mathbf{x}$ is then generated from the conditional distribution $p_\theta(\mathbf{x} \mid \mathbf{z})$ with parameters $\theta$.

In order to infer the latent activations $\mathbf{z}$ the posterior $p_\theta(\mathbf{z} \mid \mathbf{x})$ is needed. However, the true posterior $p_\theta(\mathbf{z} \mid \mathbf{x})$ is intractable, but it can be approximated by introducing the recognition model (or approximate posterior) $q_\phi(\mathbf{z} \mid \mathbf{x})$. We assume that the recognition model $q_\phi(\mathbf{z} \mid \mathbf{x})$ factorizes into the $M$ motifs and $T$ time steps of the video. In contrast to most VAE, we further assume that each latent variable $z_t^m$ is Bernoulli-distributed with parameter $\alpha_t^m(\mathbf{x}; \phi)$

$$q_\phi(\mathbf{z} \mid \mathbf{x}) = \prod_{m=1}^{M} \prod_{t=1}^{T} q_\phi(z_t^m \mid \mathbf{x}) = \prod_{m=1}^{M} \prod_{t=1}^{T} \mathrm{Bernoulli}\big(z_t^m \mid \alpha_t^m(\mathbf{x}; \phi)\big) \quad . \tag{1}$$

We sample the activations $\mathbf{z}$ in the latent space from the Bernoulli distributions to enforce sparse, sharply peaked activations. The parameters $\alpha_t^m(\mathbf{x}; \phi)$ are given by a CNN with parameters $\phi$. The corresponding plate diagram and proposed generative and recognition model are shown in figure 3.

## 3.2 THE VAE OBJECTIVE

In order to learn the variational parameters, the KL-divergence between approximate and true posterior $\mathrm{KL}(q_\phi(\mathbf{z} \mid \mathbf{x}) \| p_\theta(\mathbf{z} \mid \mathbf{x}))$ is minimized. Instead of minimizing this KL-divergence, we can also maximize the variational lower bound $\mathcal{L}(\theta, \phi; \mathbf{x})$ (ELBO) (see e.g. Blei et al. (2017))

$$\mathcal{L}(\theta, \phi; \mathbf{x}) = \mathbb{E}_{\mathbf{z} \sim q_\phi(\mathbf{z} \mid \mathbf{x})}\big[\log p_\theta(\mathbf{x} \mid \mathbf{z})\big] - \mathrm{KL}\big(q_\phi(\mathbf{z} \mid \mathbf{x}) \| p_a(\mathbf{z})\big) \quad . \tag{2}$$

In order to optimize the ELBO, the gradients w.r.t. the variational parameters $\phi$ and the generative parameters $\theta$ have to be computed. The gradient w.r.t. $\phi$, however, cannot be computed easily, since the expectation in eq. (2) depends on $\phi$. A reparameterization trick (Kingma et al., 2015) is used to overcome this problem: the random variable $\mathbf{z} \sim q_\phi(\mathbf{z} \mid \mathbf{x})$ is reparameterized using a differentiable transformation $h_\phi(\varepsilon, \mathbf{x})$ of a noise variable $\varepsilon$ such that

$$\mathbf{z} = h_\phi(\varepsilon, \mathbf{x}) \quad \text{with} \quad \varepsilon \sim p(\varepsilon) \quad . \tag{3}$$

The reparameterized ELBO, for which the expectation can be computed, e.g. using Monte Carlo sampling, is then given by

$$\mathcal{L}(\theta, \phi; \mathbf{x}) = \mathbb{E}_{\varepsilon \sim p(\varepsilon)}\big[\log p_\theta\big(\mathbf{x} \mid \mathbf{z} = h_\phi(\varepsilon, \mathbf{x})\big)\big] - \mathrm{KL}\big(q_\phi(\mathbf{z} \mid \mathbf{x}) \| p_a(\mathbf{z})\big) \quad . \tag{4}$$

More details on VAE as introduced by Kingma & Welling (2014) are given in appendix A.

## 3.3 LeMoNADe REPARAMETRIZATION TRICK AND LOSS FUNCTION

In our case, however, by sampling from Bernoulli distributions we have added discrete stochastic nodes to our computational graph, and we need to find differentiable reparameterizations of these nodes. The Bernoulli distribution can be reparameterized using the Gumbel-max trick (Luce, 1959; Yellott, 1977; Papandreou & Yuille, 2011; Hazan & Jaakkola, 2012; Maddison et al., 2014). This,

however, is not differentiable. For this reason we use the BinConcrete distribution (Maddison et al., 2016), which is a continuous relaxation of the Bernoulli distribution with temperature parameter $\lambda$. For $\lambda \to 0$ the BinConcrete distribution smoothly anneals to the Bernoulli distribution. The BinConcrete distribution can be reparameterized using the Gumbel-softmax trick (Maddison et al., 2016; Jang et al., 2017), which is differentiable.

Maddison et al. (2016) show that for a discrete random variable $\mathbf{z} \sim \text{Bernoulli}(\alpha)$, the reparameterization of the BinConcrete relaxation of this discrete distribution is

$$\tilde{\mathbf{z}} = \sigma(\mathbf{y}) = \frac{1}{1 + \exp(-\mathbf{y})} \quad \text{with} \quad \mathbf{y} = \frac{\log(\tilde{\alpha}) + \log(U) - \log(1 - U)}{\lambda} \tag{5}$$

where $U \sim \text{Uni}(0, 1)$ and $\tilde{\alpha} = \alpha/(1 - \alpha)$.

Hence the relaxed and reparameterized lower bound $\tilde{\mathcal{L}}(\theta, \tilde{\alpha}; \mathbf{x}) \approx \mathcal{L}(\theta, \phi; \mathbf{x})$ can be written as

$$\tilde{\mathcal{L}}(\theta, \tilde{\alpha}; \mathbf{x}) = \mathbb{E}_{\mathbf{y} \sim g_{\tilde{\alpha}, \lambda_1}(\mathbf{y} \,|\, \mathbf{x})} \big[ \log p_\theta \big( \mathbf{x} \,|\, \sigma(\mathbf{y}) \big) \big] - \text{KL} \big( g_{\tilde{\alpha}, \lambda_1}(\mathbf{y} \,|\, \mathbf{x}) || f_{\tilde{a}, \lambda_2}(\mathbf{y}) \big) \tag{6}$$

where $g_{\tilde{\alpha}, \lambda_1}(\mathbf{y} \,|\, \mathbf{x})$ is the reparameterized BinConcrete relaxation of the variational posterior $q_\phi(\mathbf{z} \,|\, \mathbf{x})$ and $f_{\tilde{a}, \lambda_2}(\mathbf{y})$ the reparameterized relaxation of the prior $p_a(\mathbf{z})$. $\lambda_1$ and $\lambda_2$ are the respective temperatures and $\tilde{\alpha}$ and $\tilde{a}$ the respective locations of the relaxed and reparameterized variational posterior and prior distribution.

The first term on the RHS of eq. (6) is a negative reconstruction error, showing the connection to traditional autoencoders, while the KL-divergence acts as a regularizer on the approximate posterior $q_\phi(\mathbf{z} \,|\, \mathbf{x})$. As shown in Higgins et al. (2017), we can add a $\beta$-coefficient to this KL-term which allows to vary the strength of the constraint on the latent space.

Instead of maximizing the lower bound, we will minimize the corresponding loss function

$$\ell(\mathbf{x}, \mathbf{x}', \tilde{\alpha}, \lambda_1, \tilde{a}, \lambda_2, \beta_{\text{KL}}) = \text{MSE}(\mathbf{x}, \mathbf{x}') + \beta_{\text{KL}} \cdot \text{KL} \big( g_{\tilde{\alpha}, \lambda_1}(\mathbf{y} \,|\, \mathbf{x}) || f_{\tilde{a}, \lambda_2}(\mathbf{y}) \big)$$
$$= \text{MSE}(\mathbf{x}, \mathbf{x}') - \beta_{\text{KL}} \cdot \mathbb{E}_{U \sim \text{Uni}(0,1)} \left[ \log \frac{f_{\tilde{a}, \lambda_2} \big( \mathbf{y}(U, \tilde{\alpha}, \lambda_1) \big)}{g_{\tilde{\alpha}, \lambda_1} \big( \mathbf{y}(U, \tilde{\alpha}, \lambda_1) \,|\, \mathbf{x} \big)} \right] \tag{7}$$

with $\text{MSE}(\mathbf{x}, \mathbf{x}')$ being the mean-squared error between $\mathbf{x}$ and $\mathbf{x}'$, and the $\beta$-coefficient $\beta_{\text{KL}}$. Datasets with low SNR and large background fluctuations will need a stronger regularization on the activations and hence a larger $\beta_{\text{KL}}$ than higher quality recordings. Hence, adding the $\beta$-coefficient to the loss function enables our method to adapt better to the properties of specific datasets and recording methods.

## 3.4 LeMoNADe network architecture

The encoder network starts with a few convolutional layers with small 2D filters operating on each frame of the video separately, inspired by the architecture used in Apthorpe et al. (2016) to extract cells from calcium imaging data. Afterwards the feature maps of the whole video are passed through a final convolutional layer with 3D filters. These filters have the size of the feature maps obtained from the single images times a temporal component of length $F$, which is the expected maximum temporal length of the motifs. We apply padding in the temporal domain to also capture motifs correctly which are cut off at the beginning or end of the analyzed image sequence. The output of the encoder are the parameters $\tilde{\alpha}$ which we need for the reparametrization in eq. (5). From the reparametrization we gain the activations $\mathbf{z}$ which are then passed to the decoder. The decoder consists of a single deconvolution layer with $M$ filters of the original frame size times the expected motif length $F$, enforcing the reconstructed data $\mathbf{x}'$ to be an additive mixture of the decoder filters. Hence, after minimizing the loss the filters of the decoder contain the detected motifs.

Performing these steps on the whole video would be computationally very costly. For this reason, we perform each training epoch only on a small subset of the video. The subset consists of a few hundred consecutive frames, where the starting point of this short sequence is randomly chosen in each epoch. We found that doing so did not negatively affect the performance of the algorithm. By using this strategy we are able to analyse videos of arbitrary length in a computationally efficient way.

More implementation details can be found in appendix B.

## 4 Experiments and Results

### 4.1 Synthetic data

The existence of neuronal assemblies is still fiercely debated and their detection would only be possible with automated, specifically tailored tools, like the one proposed in this paper. For this reason, no ground truth exists for the identification of spatio-temporal motifs in real neurophysiological spike data. In order to yet report quantitative accuracies, we test the algorithm on synthetically generated datasets for which ground truth is available. For the data generation we used a procedure analogous to the one used in Diego et al. (2013) and Diego & Hamprecht (2013) for testing automated pipelines for the analysis and identification of neuronal activity from calcium imaging data. In contrast to them, we include neuronal assemblies with temporal firing structure. The cells within an assembly can have multiple spikes in a randomly chosen but fixed motif of temporal length up to 30 frames. We used 3 different assemblies in each sequence. Additionally, spurious spikes of single neurons were added to simulate noise. The ratio of spurious spikes to all spikes in the dataset was varied from 0% up to 90% in ten steps. The details of the synthetic data generation can be found in appendix C.1.

To the best of our knowledge, the proposed method is the first ever to detect video motifs with temporal structure directly in calcium imaging data. As a consequence, there are no existing baselines to compare to. Hence we here propose and evaluate the SCC method presented in Peter et al. (2017) as a baseline. The SCC algorithm is able to identify motifs with temporal structure in spike trains or calcium transients. To apply it to our datasets, we first have to extract the calcium transients of the individual cells. For the synthetically generated data we know the location of each cell by construction, so this is possible with arbitrary accuracy. The output of the SCC algorithm is a matrix that contains for each cell the firing behavior over time within the motif. For a fair comparison we brought the motifs found with LeMoNADe, which are short video sequences, into the same format.

The performance of the algorithms is measured by computing the cosine similarity (Singhal, 2001) between ground truth motifs and detected motifs. The cosine similarity is one for identical and zero for orthogonal patterns. Not all ground truth motifs extend across all $30 \, \mathrm{frames}$, and may have almost vanishing luminosity in the last frames. Hence, the discovered motifs can be shifted by a few frames and still capture all relevant parts of the motifs. For this reason we computed the similarity for the motifs with all possible temporal shifts and took the maximum. More details on the computation of the similarity measure can be found in appendix C.2.

We ran both methods on 200 synthetically generated datasets with the parameters shown in table 3 in the appendix. We here show the results with the correct number of motifs ($M = 3$) used in both methods. In appendix E.1 we show that if the number of motifs is overestimated (here $M > 3$), LeMoNADe still identifies the correct motifs, but they are repeated multiple times in the surplus filters. Hence this does not reduce the performance of the algorithm. The temporal extent of the motifs was set to $F = 31$ to give the algorithms the chance to also capture the longer patterns. The cosine similarity of the found motifs to the set of ground truth motifs, averaged over all found motifs and all experiments for each of the ten noise levels, is shown in figure 4. The results in figure 4 show that LeMoNADe performs as well as SCC in detecting motifs and also shows a similar stability in the presence of noise as SCC. This is surprising since LeMoNADe does not need the previous extraction of individual cells and hence has to solve a much harder problem than SCC.

In order to verify that the results achieved by LeMoNADe and SCC range significantly above chance, we performed a bootstrap (BS) test. For this, multiple datasets were created with similar spike distributions as before, but with no reoccurring motif-like firing patterns. We compiled a distribution of similarities between patterns suggested by the proposed method and randomly sampled segments of same length and general statistics from that same BS dataset. The full BS distributions are shown in appendix C.3. The 95%-tile of the BS distributions for each noise level are also shown in figure 4.

Figure 5 shows an exemplary result from one of the analysed synthetic datasets with 10% noise and maximum temporal extend of the ground truth motifs of 28 frames. All three motifs were correctly identified (see figure 5a) with a small temporal shift. This shift does not reduce the performance as it is compensated by a corresponding shift in the activations of the motifs (see figure 5b). In order to show that the temporal structure of the found motifs matches the ground truth, in figure 5a for motif 1 and 2 we corrected the shift of one and two frames, respectively. We also show the results after

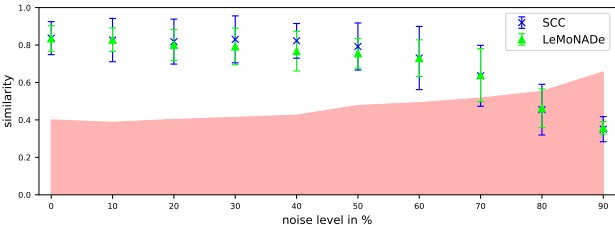

Figure 4: Similarities between found motifs and ground truth for different noise levels. We show for LeMoNADe (lime green) and SCC (blue) the average similarities between found motifs and ground truth for ten different noise levels ranging from 0% up to 90% spurious spikes. Error bars indicate the standard deviation. For each noise level 20 different datasets were analyzed. For both, LeMoNADe and SCC, the similarities between found and ground truth motifs are significantly above the 95%-tile of the corresponding bootstrap distribution (red) up to a noise level of 70% spurious spikes. Although LeMoNADe does not need the previous extraction of individual cells, it performs as well as SCC in detecting motifs and also shows a similar stability in the presence of noise.

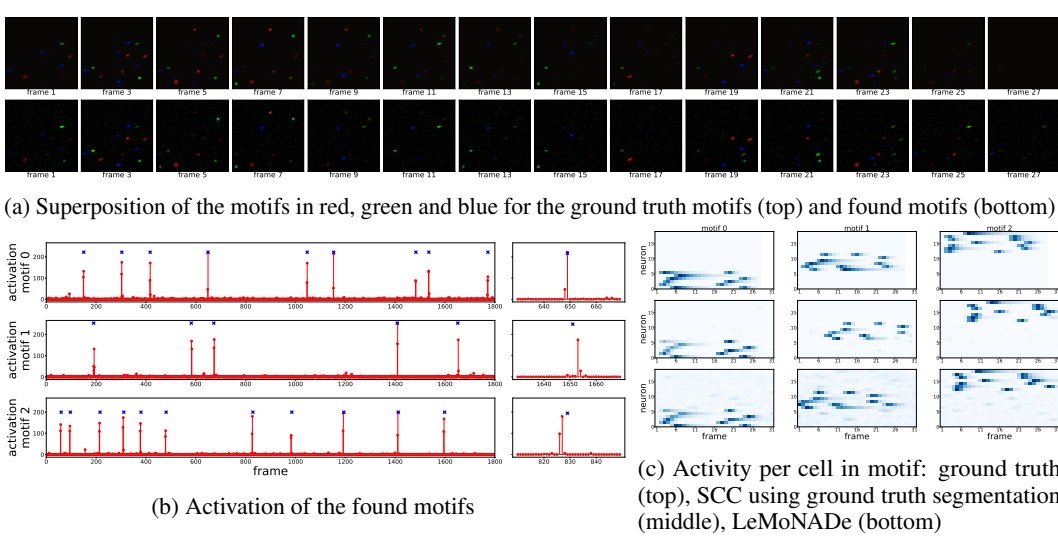

(a) Superposition of the motifs in red, green and blue for the ground truth motifs (top) and found motifs (bottom)

(b) Activation of the found motifs

(c) Activity per cell in motif: ground truth (top), SCC using ground truth segmentation (middle), LeMoNADe (bottom)

Figure 5: Exemplary result from one synthetic dataset. (a) shows a single plot containing all three motifs as additive RGB values for the ground truth motifs (top) and discovered motifs (bottom). The found motifs were ordered manually and temporally aligned to match the ground truth, for better readability. The complete motif sequences can be found in figure 10 in appendix E.1. In (b) the activations **z** of the found motifs are shown in red for the complete video (left) and a small excerpt of the sequence (right). The ground truth activations are marked with blue crosses. (c) shows the firing of the extracted cells in the ground truth motifs (top), the motifs identified by SCC (middle) and the motifs found with LeMoNADe (bottom).

extracting the individual cells from the motifs and the results from SCC in figure 5c. One can see that the results are almost identical, again except for small temporal shifts.

## 4.2 REAL DATA

We applied the proposed method on two datasets obtained from organotypic hippocampal slice cultures. The cultures were prepared from 7–9-day-old Wistar rats as described in Kann et al. (2003) and Schneider et al. (2015). The fluorescent $Ca^{2+}$ sensor, GCaMP6f (Chen et al., 2013), was delivered to the neurons by an adeno-associated virus (AAV). Neurons in stratum pyramidale of CA3 were imaged for 6.5 (dataset 1) and 5 minutes (dataset 2) in the presence of the cholinergic receptor agonist carbachol. For more details on the generation of these datasets see appendix D.1.

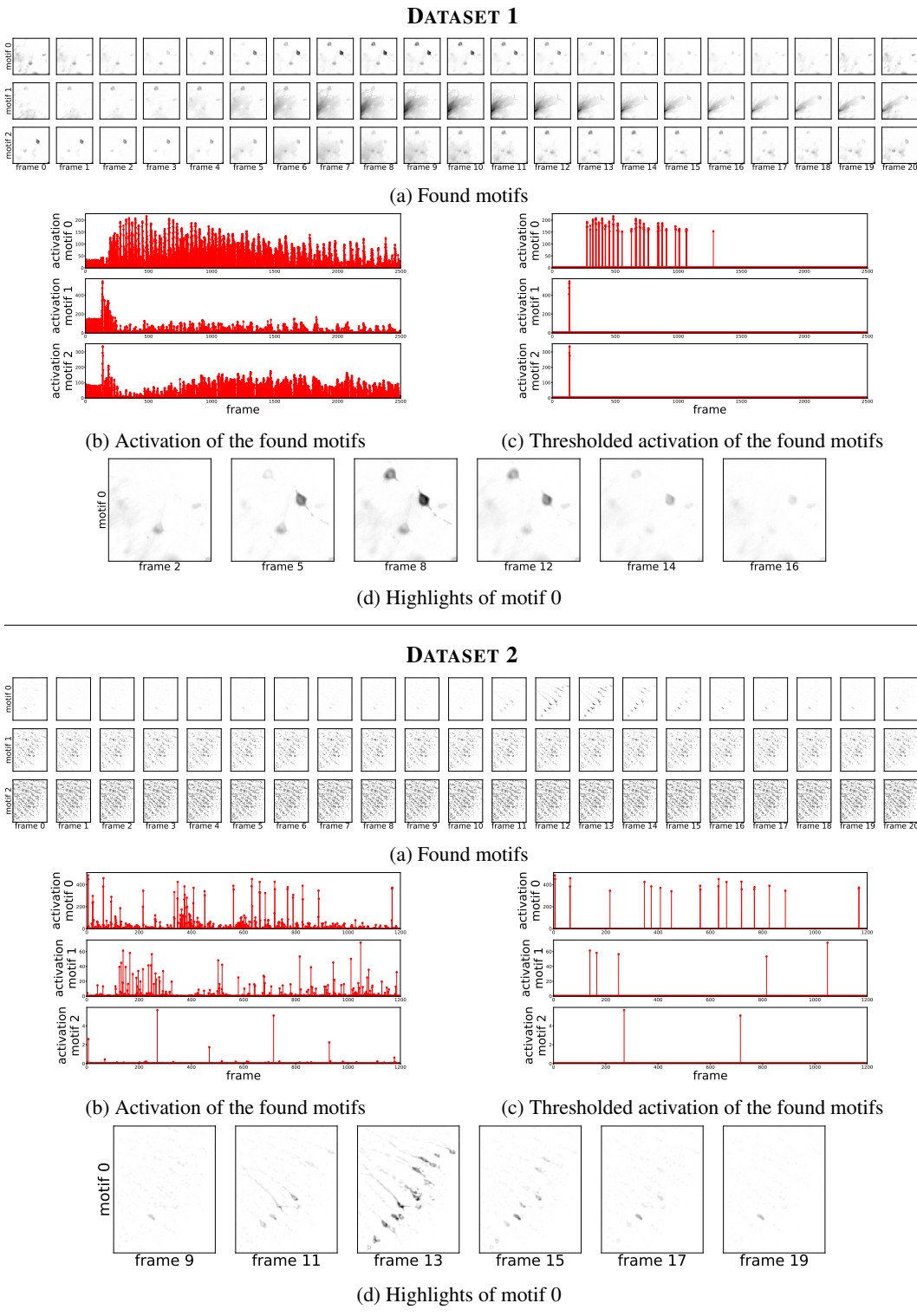

Figure 6: Result from hippocampal slice culture datasets 1 (top) and 2 (bottom). The colors in (a) are inverted compared to the standard visualization of calcium imaging data for better visibility. In (c) activations are thresholded to 70% of the maximum activation for each motif. In (d) the manually selected frames of motif 0 highlight the temporal structure of the motif.

The proposed method was run on these datasets with the parameter settings shown in table 3 in the appendix E, where we also provide additional comments on the parameter settings. The analysis of the datasets took less than two hours on a Ti 1080 GPU. Before running the analysis we computed $\Delta F/F$ for the datasets. We looked for up to three motifs with a maximum extent of $F = 21$ frames. The results are shown in figure 6. For both datasets, one motif in figure 6a consists of multiple cells, shows repeated activation over the recording period (see figure 6b, 6c), and contains temporal structure (see figure 6d). The other two "motifs" can easily be identified as artefacts and background fluctuations. As SCC and many other motif detection methods, LeMoNADe suffers from the fact that such artefacts, especially single events with extremely high neuronal activation, potentially explain a large part of the data and hence can be falsely detected as motifs. Nevertheless, these events can be easily identified by simply looking at the motif videos or thresholding the activations as done in figure 6c. Although the found motifs also include neuropil activation, this does not imply this was indeed used by the VAE as a defining feature of the motifs, just that it was also present in the images. Dendritic/axonal structures are part of the activated neurons and therefore also visible in the motif videos. If necessary, these structures can be removed by post-processing steps. As LeMoNADe reduces the problem to the short motif videos instead of the whole calcium imaging video, the neuropil subtraction becomes much more feasible.

## 5 CONCLUSION

We have presented a novel approach for the detection of neuronal assemblies that directly operates on the calcium imaging data, making the cumbersome extraction of individual cells and discrete spike times from the raw data dispensable. The motifs are extracted as short, repeating image sequences. This provides them in a very intuitive way and additionally returns information about the spatial distribution of the cells within an assembly.

The proposed method's performance in identifying motifs is equivalent to that of a state-of-the-art method that requires the previous extraction of individual cells. Moreover, we were able to identify repeating firing patterns in two datasets from hippocampal slice cultures, proving that the method is capable of handling real calcium imaging conditions.

For future work, a post-processing step as used in Peter et al. (2017) or a group sparsity regularization similar to the ones used in Bascol et al. (2016) or Mackevicius et al. (2018) could be added to determine a plausible number of motifs automatically. Moreover, additional latent dimensions could be introduced to capture artefacts and background fluctuations and hence automatically separate them from the actual motifs. The method is expected to, in principle, also work on other functional imaging modalities. We will investigate the possibility of detecting motifs using LeMoNADe on recordings from human fMRI or voltage-sensitive dyes in the future.

### ACKNOWLEDGMENTS

EK thanks Ferran Diego for sharing his knowledge on generating synthetic data and for his scientific advice. DD acknowledges partial financial support by DFG Du 354/8-1. EK, HS, JS, SE, OK, DD and FAH gratefully acknowledge partial financial support by DFG SFB 1134.

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

# APPENDIX

## A   VARIATIONAL AUTOENCODER

Variational autoencoder (VAE) are generative latent variable models which were first described in Kingma & Welling (2014). The data $\mathbf{x} = \{\mathbf{x}^{(i)}\}_{i=1}^{N}$, consisting of $N$ samples of some random variable $\mathbf{x}$, is generated by first drawing a latent variable $\mathbf{z}^{(i)}$ from a prior distribution $p(\mathbf{z})$ and then sampling from the conditional distribution $p_{\theta*}(\mathbf{x}\,|\,\mathbf{z})$ with parameters $\theta^*$. The distribution $p_{\theta*}(\mathbf{x}\,|\,\mathbf{z})$ belongs to the parametric family $p_\theta(\mathbf{x}\,|\,\mathbf{z})$ with differentiable PDFs w.r.t. $\theta$ and $\mathbf{z}$. Both the true parameters $\theta^*$ as well as the latent variables $\mathbf{z}^{(i)}$ are unknown. We are interested in an approximate posterior inference of the latent variables $\mathbf{z}$ given some data $\mathbf{x}$. The true posterior $p_\theta(\mathbf{z}\,|\,\mathbf{x})$, however, is usually intractable. But it can be approximated by introducing the recognition model (or approximate posterior) $q_\phi(\mathbf{z}\,|\,\mathbf{x})$. We want to learn both the recognition model parameters $\phi$ as well as the generative model parameters $\theta$. The recognition model is usually referred to as the probabilistic encoder and $p_\theta(\mathbf{x}\,|\,\mathbf{z})$ is called the probabilistic decoder.

In order to learn the variational parameters $\phi$ we want to minimise the KL-divergence between approximate and true posterior $\mathrm{KL}(q_\phi(\mathbf{z}|\mathbf{x})\|p_\theta(\mathbf{z}|\mathbf{x}))$. Therefore we use the fact that the marginal likelihood $p_\theta(\mathbf{x})$ can be written as

$$\log p_\theta(\mathbf{x}) = \mathcal{L}(p, q; \mathbf{x}) + \mathrm{KL}\big(q_\phi(\mathbf{z}|\mathbf{x})\|p_\theta(\mathbf{z}|\mathbf{x})\big) \tag{8}$$

As the KL-divergence is non-negative, we can minimize $\mathrm{KL}\big(q_\phi(\mathbf{z}|\mathbf{x})\|p_\theta(\mathbf{z}|\mathbf{x})\big)$ by maximizing the (variational) lower bound $\mathcal{L}(p, q; \mathbf{x})$ with

$$\mathcal{L}(p, q; \mathbf{x}) = \mathbb{E}_{\mathbf{z}\sim q_\phi(\mathbf{z}|\mathbf{x})}\left[\log p_\theta(\mathbf{x}|\mathbf{z})\right] - \mathrm{KL}\big(q_\phi(\mathbf{z}|\mathbf{x})\|p(\mathbf{z})\big) \quad . \tag{9}$$

In order to optimise the lower bound $\mathcal{L}(p, q; \mathbf{x})$ w.r.t. both the variational parameters $\phi$ and the generative parameters $\theta$, we need to compute the gradients

$$\nabla_{\phi,\theta}\mathcal{L}(p, q; \mathbf{x}) = \nabla_{\phi,\theta}\mathbb{E}_{\mathbf{z}\sim q_\phi(\mathbf{z}|\mathbf{x})}\left[\log p_\theta(\mathbf{x}|\mathbf{z})\right] - \nabla_{\phi,\theta}\mathrm{KL}\big(q_\phi(\mathbf{z}|\mathbf{x})\|p(\mathbf{z})\big) \quad . \tag{10}$$

For the first part of the lower bound the gradient w.r.t. $\theta$ can be easily computed using Monte Carlo sampling

$$\nabla_\theta\mathbb{E}_{\mathbf{z}\sim q_\phi(\mathbf{z}|\mathbf{x})}\left[\log p_\theta(\mathbf{x}|\mathbf{z})\right] = \mathbb{E}_{\mathbf{z}\sim q_\phi(\mathbf{z}|\mathbf{x})}\left[\nabla_\theta\log p_\theta(\mathbf{x}|\mathbf{z})\right] \approx \frac{1}{S}\sum_{s=1}^{S}\nabla_\theta\log p_\theta(\mathbf{x}|\mathbf{z}^s) \tag{11}$$

with $\mathbf{z}^s \sim q_\phi(\mathbf{z}|\mathbf{x})$. The gradient w.r.t. $\phi$, however, does not take the form of an expectation in $\mathbf{z}$ and can therefore not be sampled that easily:

$$\nabla_\phi\mathbb{E}_{\mathbf{z}\sim q_\phi(\mathbf{z}|\mathbf{x})}\left[\log p_\theta(\mathbf{x}|\mathbf{z})\right] = \nabla_\phi\int q_\phi(\mathbf{z}|\mathbf{x})\log p_\theta(\mathbf{x}|\mathbf{z})d\mathbf{z} = \int \log p_\theta(\mathbf{x}|\mathbf{z})\nabla_\phi q_\phi(\mathbf{z}|\mathbf{x})d\mathbf{z} \quad . \tag{12}$$

However, in most cases we can use the reparameterization trick to overcome this problem: the random variable $\tilde{\mathbf{z}} \sim q_\phi(\mathbf{z}\,|\,\mathbf{x})$ can be reparameterised using a differentiable transformation $h_\phi(\varepsilon, \mathbf{x})$ of a noise variable $\varepsilon$ such that

$$\tilde{\mathbf{z}} = h_\phi(\varepsilon, \mathbf{x}) \quad \text{with} \quad \varepsilon \sim p(\varepsilon) \tag{13}$$

We now can compute the gradient w.r.t. $\phi$ again using Monte Carlo sampling

$$\nabla_\phi\mathbb{E}_{\varepsilon\sim p(\varepsilon)}\left[\log p_\theta(\mathbf{x}|\mathbf{z} = h_\phi(\varepsilon, \mathbf{x}))\right] = \mathbb{E}_{\varepsilon\sim p(\varepsilon)}\left[\nabla_\phi\log p_\theta(\mathbf{x}|\mathbf{z} = h_\phi(\varepsilon, \mathbf{x}))\right]$$

$$\approx \frac{1}{S}\sum_{s=1}^{S}\nabla_\phi\log p_\theta(\mathbf{x}|\mathbf{z}^s = h_\phi(\varepsilon^s, \mathbf{x})) \tag{14}$$

with $\varepsilon^s \sim p(\varepsilon)$. Hence, the reparameterized lower bound $\tilde{\mathcal{L}}(p, q; \mathbf{x}) \approx \mathcal{L}(p, q; \mathbf{x})$ can be written as

$$\tilde{\mathcal{L}}(p, q; \mathbf{x}) = \frac{1}{S}\sum_{s=1}^{S}\log p_\theta(\mathbf{x}|\mathbf{z}^s) - \mathrm{KL}(q_\phi(\mathbf{z}\,|\,\mathbf{x})\|p(\mathbf{z})) \tag{15}$$

with $\mathbf{z}^s = h_\phi(\varepsilon^s, \mathbf{x}), \varepsilon \sim p(\varepsilon)$. The first term on the RHS of eq. (15) is a negative reconstruction error, showing the connection to traditional autoencoders, while the KL-divergence acts as a regularizer on the approximate posterior $q_\phi(\mathbf{z}\,|\,\mathbf{x})$.

## B LeMoNADe NETWORK ARCHITECTURE AND IMPLEMENTATION DETAILS

### B.1 ENCODER

The encoder network starts with a few convolutional layers with small 2D filters operating on each frame of the video separately, inspired by the architecture used in Apthorpe et al. (2016) to extract cells from calcium imaging data. The details of this network are shown in table 1. Afterwards the feature maps of the whole video are passed through a final convolutional layer with 3D filters. These filters have size of the feature maps gained from the single images times a temporal component of length $F$, which is the expected maximum temporal extent of a motif. We use $2 \cdot M$ filters and apply padding in the temporal domain to avoid edge effects. By this also motifs that are cut off at the beginning or the end of the sequence can be captured properly. The output of the encoder are $2 \cdot M$ feature maps of size $(T + F - 1) \times 1 \times 1$.

### B.2 REPARAMETERIZATION

Instead of reparameterizing the Bernoulli distributions, we will reparameterize their BinConcrete relaxations. The BinConcrete relaxation of a Bernoulli distribution with parameter $\alpha$ takes as input parameter $\tilde{\alpha} = \alpha/(1 - \alpha)$. Maddison et al. (2016) showed that instead of using the normalized probabilities $\alpha$, we can also perform the reparametrization with unnormalized parameters $\alpha^1$ and $\alpha^2$, where $\alpha^1$ is the probability to sample a one and $\alpha^2$ is the probability to sample a zero and $\tilde{\alpha} = \alpha^1/\alpha^2$.

The first $M$ feature maps, which were outputted by the encoder, are assigned to contain the unnormalised probabilities $\alpha^1_{m,t}$ for the activation of motif $m$ in frame $t$ to be one. The second $M$ feature maps contain the unnormalized probabilities $\alpha^2_{m,t}$ for the activation of motif $m$ in frame $t$ to be zero. The parameter $\tilde{\alpha}$ that is needed for the reparameterized BinConcrete distribution is obtained by dividing the two vectors elementwise: $\tilde{\alpha}^m_t = \alpha^1_{m,t}/\alpha^2_{m,t}$. We use the reparameterization trick to sample from BinConcrete($\tilde{\alpha}^m_t$) as follows: First we sample $\left\{ \{U^m_t\}_{t=1}^{T+F-1} \right\}_{m=1}^{M}$ from a uniform distribution Uni$(0, 1)$. Next, we compute $\mathbf{y}$ with

$$y^m_t = \left( \frac{\tilde{\alpha}^m_t \cdot U^m_t}{1 - U^m_t} \right)^{1/\lambda_1} \quad . \tag{16}$$

Finally, we gain $\mathbf{z}$ according to

$$z^m_t = \frac{y^m_t}{1 + y^m_t} \cdot \alpha^1_{m,t} \tag{17}$$

for all $m = 1, \ldots, M$ and $t = 1, \ldots, T + F - 1$. The multiplication by $\alpha^1_{m,t}$ in eq. (17) is not part of the original reparametrization trick (Maddison et al., 2016; Jang et al., 2017). But we found that the results of the algorithm improved dramatically as we scaled the activations with the $\alpha^1$-values that were originally predicted from the encoder network.

### B.3 DECODER

The input to the decoder are now the activations $\mathbf{z}$. The decoder consists of a single deconvolution layer with $M$ filters of the original frame size times the expected motif length $F$. These deconvolution filters contain the motifs we are looking for.

The details of the used networks as well as the sizes of the inputs and outputs of the different steps are shown in table 1. Algorithm 1 summarizes the reparametrization and updates.

## C EXPERIMENTS AND RESULTS ON SYNTHETIC DATA

### C.1 SYNTHETIC DATA GENERATION

We created 200 artificial sequences of length $60\,\mathrm{s}$ with a frame rate of $30\,\mathrm{fps}$ and $128 \times 128\,\mathrm{pixel}$ per image. The number of cells was varied and they were located randomly in the image plane with an overlap of up to $30\,\%$. The cell shapes were selected randomly from 36 shapes extracted from

Table 1: LeMoNADe network architecture details

| Operation | Kernel | Feature maps | Padding | Stride | Nonlinearity |
|---|---|---|---|---|---|
| | | Input: $T$ images, $P \times P'$ | | | |
| 2D Convolution | $3 \times 3$ | 24 | $0 \times 0$ | 1 | ELU |
| 2D Convolution | $3 \times 3$ | 48 | $0 \times 0$ | 1 | ELU |
| Max-Pooling | $2 \times 2$ | – | $0 \times 0$ | 2 | – |
| 2D Convolution | $3 \times 3$ | 72 | $0 \times 0$ | 1 | ELU |
| 2D Convolution | $3 \times 3$ | 96 | $0 \times 0$ | 1 | ELU |
| Max-Pooling | $2 \times 2$ | – | $0 \times 0$ | 2 | – |
| 2D Convolution | $3 \times 3$ | 120 | $0 \times 0$ | 1 | ELU |
| 2D Convolution | $1 \times 1$ | 48 | $0 \times 0$ | 1 | ELU |
| | | Output: $T$ images, $\tilde{P} \times \tilde{P}'$, $\tilde{P} = ((P-4)/2 - 4)/2 - 2$, $\tilde{P}' = ((P'-4)/2 - 4)/2 - 2$ | | | |
| | | | | | |
| | | Input: 1 video, $T \times \tilde{P} \times \tilde{P}'$ | | | |
| 3D Convolution | $F \times \tilde{P} \times \tilde{P}'$ | $2M$ | $(F-1) \times 0 \times 0$ | 1 | SoftPlus |
| | | Output: $2M$ feature maps, $(T + F - 1) \times 1 \times 1$ | | | |
| | | | | | |
| | | Input: $2M$ feature maps, $(T + F - 1) \times 1 \times 1$ | | | |
| Reparametrization | – | – | – | – | – |
| | | Output: $M$ activations, $(T + F - 1) \times 1 \times 1$ | | | |
| | | | | | |
| | | Input: $M$ activations, $(T + F - 1) \times 1 \times 1$ | | | |
| 3D TransposedConvolution | $F \times P \times P'$ | $M$ | $(F-1) \times 0 \times 0$ | 1 | ReLU |
| | | Output: 1 video, $T \times P \times P'$ | | | |

---

**Algorithm 1:** The LeMoNADe algorithm

**Input:** raw video $\mathbf{x}$, normalized to zero mean and unit variance, architectures $f_\theta, \alpha_\phi$,
      hyperparameter $\lambda_1, \lambda_2, \tilde{a}, \beta_{\text{KL}}$
**Result:** trained $f_\theta, \alpha_\phi$

$\theta, \phi \leftarrow$ Initialize network parameters
**repeat**
    // Sample subset of video
    $\mathbf{x}_{\text{sub}} \leftarrow$ Randomly chosen sequence of consecutive frames from $\mathbf{x}$
    // Encoding step
    Encode $\mathbf{x}_{\text{sub}}$ to get $\tilde{\alpha}$ as described in section B.1 and B.2
    // Latent Step
    Sample noise $U \sim \text{Uni}(0,1)$
    Compute $\mathbf{y}$ following eq. (16)
    Compute $\mathbf{z}$ following eq. (17)
    // Decoding Step
    $\mathbf{x}'_{\text{sub}} \leftarrow$ decode via $f_\theta(\mathbf{z})$
    // Update Parameters
    Compute gradients of loss
    $\phi, \theta \leftarrow$ update via $\nabla_{\phi,\theta} \ell(\mathbf{x}_{\text{sub}}, \mathbf{x}'_{\text{sub}}, \tilde{\alpha}, \lambda_1, \tilde{a}, \lambda_2, \beta_{\text{KL}})$ (see eq. (7) in the main paper)
**until** *until convergence of $\theta, \phi$;*

---

real data. The transients were modelled as two-sided exponential decay with scales of $50\,\text{ms}$ and $400\,\text{ms}$, respectively. In contrast to Diego & Hamprecht (2013), we included neuronal assemblies with temporal firing structure. That means cells within an assembly can perform multiple spikes in a randomly chosen but fixed motif of temporal length up to 30 frames. We used 3 different assemblies in each sequence. The assembly activity itself was modelled as a Poisson process (Lopes-dos Santos et al., 2013) with a mean of $0.15\,\text{spikes/second}$ and a refractory period of at least the length of the motif itself. By construction the cell locations as well as the firing motifs are known for these datasets. In order to simulate the conditions in real calcium imaging videos as good as possible, we added Gaussian background noise with a relative amplitude $(\text{max intensity} - \text{mean intensity})/\sigma_{\text{noise}}$ between 10 and 20. Additionally, spurious spikes not belonging to any motif were added. The amount

Table 2: Average cosine similarity between ground truth and discovered motifs. The average similarity together with the standard deviation were computed over 20 different datasets for each noise level, both for LeMoNADe and SCC. A bootstrap distribution of similarities was computed (see section C.3). BS-95 gives the 5% significance threshold of this distribution.

| | on video data | | after cell extraction |
|---|---|---|---|
| NOISE LEVEL | **LeMoNADe** | BS-95 | SCC |
| 0% | $\mathbf{0.838 \pm 0.066}$ | 0.400 | $0.837 \pm 0.088$ |
| 10% | $0.826 \pm 0.061$ | 0.387 | $0.826 \pm 0.116$ |
| 20% | $0.804 \pm 0.080$ | 0.402 | $0.818 \pm 0.120$ |
| 30% | $0.770 \pm 0.130$ | 0.413 | $0.830 \pm 0.125$ |
| 40% | $0.775 \pm 0.107$ | 0.426 | $0.822 \pm 0.093$ |
| 50% | $0.756 \pm 0.079$ | 0.477 | $0.791 \pm 0.126$ |
| 60% | $0.730 \pm 0.098$ | 0.492 | $0.731 \pm 0.169$ |
| 70% | $0.639 \pm 0.142$ | 0.516 | $0.636 \pm 0.163$ |
| 80% | $0.462 \pm 0.103$ | 0.553 | $0.454 \pm 0.135$ |
| 90% | $0.357 \pm 0.034$ | 0.656 | $0.351 \pm 0.067$ |

of spurious spikes was varied from 0% up to 90% of all spikes in the dataset. For each of the 10 noise levels 20 datasets were generated.

## C.2 SIMILARITY MEASURE

The performance of the algorithms is measured by computing the cosine similarity (Singhal, 2001) between ground truth motifs and found motifs. The found motifs are in an arbitrary order, not necessarily corresponding to the order of the ground truth motifs. Additionally, the found motifs can be shifted in time compared to the ground truth. To account for this fact, we compute the similarity between the found motifs and each of the ground truth motifs with all possible temporal shifts and take the maximum. Hence, the similarity between the $m$-th found motif and the set of ground truth motifs $\mathcal{G}$ is defined by

$$Sim(\mathcal{M}^m, \mathcal{G}) = \max \left\{ \frac{\langle \text{vec}(\mathcal{M}^m), \text{vec}(\overset{s\rightarrow}{G}) \rangle}{\|\text{vec}(\mathcal{M}^m)\|_2 \cdot \|\text{vec}(\overset{s\rightarrow}{G})\|_2} \,\Big|\, G \in \mathcal{G}, s \in \{-F, \ldots, F\} \right\} \quad (18)$$

where $\mathcal{M}^m$ is the $m$-th found motif, $\langle \cdot, \cdot \rangle$ is the dot product and $\text{vec}(\cdot)$ vectorizes the motifs with dimensions $F \times N$ into a vector of length $F \cdot N$, where $N$ is the number of cells. The shift operator $\overset{s\rightarrow}{(\cdot)}$ moves a motif $s$ frames forward in time while keeping the same size and filling missing values appropriately with zeros (Smaragdis, 2004).

The cosine similarity of the found motifs to the set of ground truth motifs was averaged over all found motifs and all experiments for each noise level. The average similarities achieved with LeMoNADe and SCC as well as the 5% significance threshold of the BS distribution for each noise level can be found in table 2.

## C.3 BOOTSTRAP-BASED SIGNIFICANCE TEST

Statistical methods for testing for cell assemblies (or spatio-temporal patterns more generally) have been advanced tremendously in recent years, addressing many of the issues that have plagued older approaches (Grün, 2009; Staude et al., 2010a;b; Russo & Durstewitz, 2017). Simple shuffle bootstraps are not necessarily the best methods if they destroy too much of the auto-correlative structure, and they can severely underestimate the distributional tails (Davison et al., 1997). Therefore we use sophisticated parametric, model-based bootstraps which retain the full statistical structure of the original data, except for the crucial feature of repeating motifs.

In order to provide a 'null hypothesis (H0)' reference for the motif similarities returned by LeMoN-ADe (or other methods), we used the following bootstrap (BS) based test procedure: We generated 20 datasets analogue to those described in section C.1, i.e. with same spiking statistics and temporal

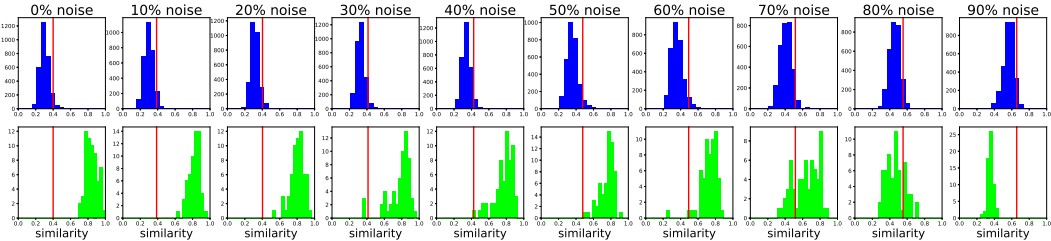

Figure 7: Top: Bootstrap distribution for similarity between random patterns. Shown is a sample from the BS distribution (blue) and the 95% significance threshold (red). Bottom: Distribution for similarity between patterns found on data which contained repeating motifs. Shown are the similarities between motifs found with LeMoNADe (lime green) and the ground truth motifs for the synthetic datasets discussed in the paper, which contained repeating motifs. The 95% significance threshold of the corresponding BS distribution is indicated as vertical red line.

convolution with calcium transients, but without repeating motifs. These motif-less H0 datasets were then processed by LeMoNADe in the very same way as the motif-containing datasets, i.e. with the parameter settings as shown in table 3. From each of these BS datasets 150 random samples of the same temporal length as that of the 'detected' motifs were drawn. For each BS dataset, the similarities between each of the found motifs and all of the 150 random samples were computed as described in section C.2. As datasets with higher noise levels have different spiking statistics, we repeated this procedure for each of the ten noise levels.

Figure 7 shows the BS distributions (top). We also show the distribution of similarities between motifs found with LeMoNADe on the datasets which contained motifs (bottom). The 95%-tile (corresponding to a 5% alpha level) of the BS distribution is displayed as vertical red line. Up to a noise level of 70% the average of the similarities found on the datasets that contained motifs is much higher than the 95%-tile of the BS distribution.

# D  EXPERIMENTS AND RESULTS ON REAL DATA

## D.1  DATA GENERATION

Organotypic hippocampal slice cultures were prepared from 7–9-day-old Wistar rats (Charles River Laboratories, Sulzfeld, Germany) as described by Kann et al. (2003) and Schneider et al. (2015). Animals were taken care of and handled in accordance with the European directive 2010/63/EU and with consent of the animal welfare officers at Heidelberg University (license, T96/15).

Slices were infected with adeno-associated virus (AAV) obtained from Penn Vector Core (PA, USA) encoding GCaMP6f under the control of the CamKII promoter AAV5.CamKII.GCaMPf.WPRE.SV40, Lot # V5392MI-S). AAV transduction was achieved, under sterile conditions, by applying $0.5\mu l$ of the viral particles solution (qTiter: 1.55e13 GC/ml) on top of the slices.

Slices were maintained on Biopore membranes (Millicell standing inserts; Merck Millipore, Schwalbach, Germany) between culture medium. The medium consisted of 50% minimal essential medium, 25% Hank's balanced salt solution (Sigma-Aldrich, Taufkirchen, Germany), 25% horse serum (Life Technologies, Darmstadt, Germany), and $2mM$ L-glutamine (Life Technologie) at pH 7.3, stored in an incubator (Heracell; Thermoscientific, Dreieich, Germany) with humidified normal atmosphere (5% $CO_2$, $36.5°C$). The culture medium (1 ml) was replaced three times per week.

Artificial cerebrospinal fluid used for imaging was composed of 129 mM NaCl, 3 mM KCl, 1.25 mM NaH2PO4, 1.8 mM MgSO4, 1.6 mM CaCl2, 21 mM NaHCO3, and10 mM glucose (Sigma-Aldrich, Taufkirchen, Germany). The pH of the recording solution was 7.3 when it was saturated with the gas mixture (95% O2, 5% CO2). Recording temperature was $32 \pm 1°C$. Constant bath wash of $20\mu M$ (dataset 1) and $10\mu M$ (dataset 2) carbachol (Sigma-Aldrich) was performed to enhance neuronal activity and increase firing probability during imaging (Müller et al., 1988).

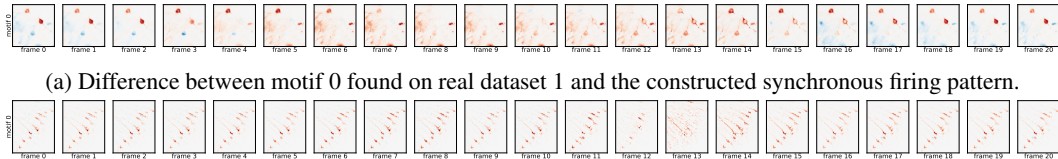

(a) Difference between motif 0 found on real dataset 1 and the constructed synchronous firing pattern.

(b) Difference between motif 0 found on real dataset 2 and the constructed synchronous firing pattern.

Figure 8: Color-coded difference between discovered motifs and intensity modulated synchronous firing. Red color indicates negative differences, blue positive differences and white zero difference. The fact that for both datasets in motif 0 some cells are displayed in red over multiple frames shows that these motifs contain temporal structure beyond mere spiking synchrony.

Imaging of CA3 region of the hippocampus was performed on day 29 with 20x magnification (dataset 1) and on day 30 with 10x magnification (dataset 2) in vitro (23 days post viral infection) from slices maintained in submerged chamber of Olympus BX51WI microscope. GCaMP6f was excited at $485 \pm 10nm$. Fluorescence images (emission at $521 \pm 10nm$) were recorded at $6.4Hz$ (dataset 1) and $4Hz$ (dataset 2) using a CCD camera (ORCA-ER; Hamamatsu Photonics, Hamamatsu City, Japan). Before running the analysis we computed $\Delta F/F$ for the datasets. In order to perform the computations more efficiently, we cropped the outer parts of the images containing no interesting neuronal activity and downsampled dataset 2 by a factor of $0.4$.

### D.2 TEMPORAL STRUCTURE PLOTS

In order to show that the motifs 0 found in the two real datasets contain temporal structure, we compare them to what the synchronous activity of the participating cells with modulated amplitude would look like. The synchronous firing pattern was constructed as follows: First, for the motif $\mathcal{M}^m$ with $m = 1, \ldots, M$ the maximum projection $\mathcal{P}^m$ at each pixel $p = 1, \ldots, P \cdot P'$ over time was computed by

$$\mathcal{P}_p^m = \max_f \mathcal{M}_p^m \quad \text{with } f = 1, \ldots, F \tag{19}$$

and normalized

$$\tilde{\mathcal{P}}_p^m = \frac{\mathcal{P}_p^m}{\max_p' \mathcal{P}^m} \quad . \tag{20}$$

Finally, the synchronous firing pattern $\mathcal{S}^m$ for motif $m$ is gained by multiplying this normalized maximum projection at each time frame $f$ with the maximum intensity of motif $m$ at that frame:

$$\mathcal{S}_f^m = \tilde{\mathcal{P}}^m \cdot \max_p \mathcal{M}_f^m \quad \text{for } f = 1, \ldots, F \quad . \tag{21}$$

Figures 8 shows the difference between the found motif and the constructed synchronous firing patterns for the motifs found on the two real datasets.

### D.3 COMPARISON TO RESULTS OBTAINED WITH SCC

In order to show that LeMoNADe performs similar to SCC not only on synthetically generated data but also on real data, we ran both methods on real dataset 2. A well trained neuroscientist manually extracted the individual cells and calcium traces from the original calcium imaging video. Figure 9a shows the result obtained with SCC on these traces. In the same manner calcium traces were extraced from the motif found with LeMoNADe (see figure 9b). Both results in figure 9 are highly similar.

### E PARAMETER SETTINGS

LeMoNADe is not more difficult to apply than other motif detection methods for neuronal spike data. In our experiments, for most of the parameters the default settings worked well on different datasets and only three parameters need to be adjusted: the maximum number of motifs $M$, the maximum

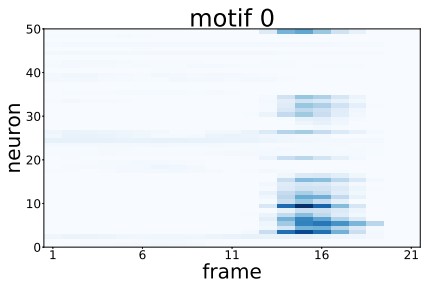
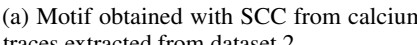
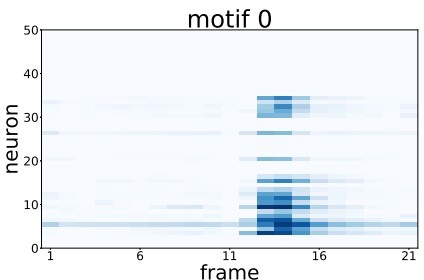

(a) Motif obtained with SCC from calcium traces extracted from dataset 2.

(b) Traces obtained from the motif found with LeMoNADe on dataset 2.

Figure 9: Result obtained on the real dataset 2 after manual cell extraction with SCC (a) and the traces manually extracted from the motif found with LeMoNADe on the original video data (b).

Table 3: Parameters used for the shown experiments. $M$ is the number of motifs, $F$ the maximum temporal extent of a motif, $\lambda_1$ and $\lambda_2$ are the temperatures for the relaxed approximate posterior and prior distributions, $\tilde{a}$ is the location of the BinConcrete prior, $b$ is the number of consecutive frames analysed in each epoch, and $\beta_{\text{KL}}$ is the weight of the KL-regularization term in the loss function. $\beta_e$ is the ensemble-penalty used in SCC.

| | $M$ | $F$ | $\tilde{a}$ | $\lambda_1$ | $\lambda_2$ | #epochs | learning rate | $b$ | $\beta_{\text{KL}}$ |
|---|---|---|---|---|---|---|---|---|---|
| LeMoNADe on synth. datasets with noise level $< 50\%$ | 3 | 31 | 0.05 | 0.6 | 0.5 | 5000 | $10^{-5}$ | 500 | 0.10 |
| LeMoNADe on synth. datasets with noise level $\geq 50\%$ | 3 | 31 | 0.10 | 0.6 | 0.5 | 5000 | $10^{-5}$ | 500 | 0.10 |
| LeMoNADe on real dataset 1 | 3 | 21 | 0.05 | 0.4 | 0.3 | 5000 | $10^{-5}$ | 150 | 0.01 |
| LeMoNADe on real dataset 2 | 3 | 21 | 0.01 | 0.6 | 0.5 | 5000 | $10^{-5}$ | 500 | 0.10 |

| | $M$ | $F$ | $\beta_e$ | | | #epochs | #inits | | |
|---|---|---|---|---|---|---|---|---|---|
| SCC on synth. datasets | 3 | 31 | $10^{-4}$ | | | 10 | 1 | | |

motif length $F$, and one of the sparsity parameters (e.g. $\tilde{a}$ or $\beta_{\text{KL}}$). For SCC the user also has to specify three similar parameters. In addition, SCC requires the previous extraction of a spike matrix which implies many additional parameters.

Table 3 shows the parameter settings used for the experiments shown in the paper.

### E.1 OVER- AND UNDER-ESTIMATION OF THE MAXIMUM NUMBER OF MOTIFS $M$

In order to show the effects of over- and underestimating the number of motifs, we first use our synthetic data with existing ground truth and 3 true motifs and run LeMoNADe with underestimated ($M = 1$), correct ($M = 3$) and overestimated ($M = 5$) number of expected motifs. Figure 10 shows the complete ground truth (figure 10a) and found motifs for the exemplary synthetic dataset discussed in the paper. Besides the results for $M = 3$ (figure 10c) we also show the found motifs for $M = 1$ (figure 10b) and $M = 5$ (figure 10d). If the number of motifs is underestimated ($M = 1$) only one of the true motifs is captured. When the number of motifs is overestimated ($M = 5$) the correct motifs are identified and the surplus filters are filled with (shifted) copies of the true motifs and background noise.

We also investigated the influence of different numbers of motifs on the results on real datasets. Figure 11 shows the found motifs on dataset 1 for the different numbers of motifs $M = 1, 2, 3, 5$. When the number is limited (as for $M = 1$), the model is expected to learn those motifs first which best explain the data. The motif shown in figure 11a also appears if $M$ is increased. This shows that this motif is highly present in the data. However, as long as only one filter is available the motif also contains a lot of background noise. The second filter in figure 11b contains a high luminosity artefact of the data. With its high luminosity and large spacial extent, it explains a lot of the dataset. However, it can easily be identified as no neuronal assembly. If the number of motifs is further increased to $M = 3$ (see figure 11c), more background noise is captured in the additional filter and the motif becomes cleaner. When the number of motifs is further increased to $M = 5$, no new motifs appear

and the surplus two filters seem to be filled up with parts of the structures which were already present in 11c.

Hence, when the correct number of motifs is unknown (as expected for real datasets) we recommend to slightly overestimate the expected number of motifs. The result will capture the true motifs plus some copies of them. In future work, a post-processing step as in Peter et al. (2017) or a group sparsity regularization as in Bascol et al. (2016) and Mackevicius et al. (2018) could be introduced to eliminate these additional copies automatically. Background noise could be easily identified as no motif by either looking at the motif videos or thresholding the found activations. In future extends of the model we will study the effect of additional latent dimensions for background noise to automatically separate it from actual motifs.

### E.2 OVER- AND UNDER-ESTIMATION OF THE MAXIMUM MOTIF LENGTH $F$

If the maximum motif length $F$ is underestimated the found motifs are expected to just contain the part of the motif that reduces the reconstruction error most. Hence in most cases the most interesting parts of the motifs will be captured but details at either end of the motifs could be lost. If the motif length is overestimated, the motifs can be captured completely but might be shifted in time. This shift, however, will be compensated by the motif activations and hence has no negative effect on the results. In our experiments we achieved good results with a generously chosen motif length. For this reason we recommend to overestimate the motif length.

Figure 12 shows the found motifs on real dataset 1 with $M = 3$ and for the different motif lengths $F = 21$ and $F = 31$. The results are highly similar. In both cases, the interesting pattern (motif 0 in figure 12a and motif 1 in figure 12b, respectively) is captured.

### E.3 SPARSITY PARAMETER

The parameter $\tilde{a}$ influences the sparsity of the found activations. Smaller values of $\tilde{a}$ will penalize activations harder and hence often result in cleaner and more meaningful motifs. However, if $\tilde{a}$ is too small it will suppress the activations completely. For this reason we recommend to perform for each new dataset experiments with different values of $\tilde{a}$. Changing the value of $\beta_{\mathrm{KL}}$ is another option to regulate the sparsity of the activations. However, in our experiments we found that the default value of $\beta_{\mathrm{KL}} = 0.1$ worked well for many different datasets and varying $\tilde{a}$ was effective enough. For the temperature parameters the default values $\lambda_1 = 0.6$ and $\lambda_2 = 0.5$ worked well in most cases and changing them is usually not necessary.

In order to show the reaction of the method to the choice of $\tilde{a}$ and $\beta_{\mathrm{KL}}$ we performed multiple experiments on the real dataset 2 with different parameter settings. We fixed all parameters as shown in table 3 except for $\tilde{a}$ (figures 13 and 14) and $\beta_{\mathrm{KL}}$ (figures 15 and 16).

When $\tilde{a}$ is varied within one order of magnitude (see figure 13) the motifs look quite similar - except for temporal shifts of the motifs and shuffling of the order of the motifs. For smaller values of $\tilde{a}$ surplus filters are filled with background noise (see figures 13a to 13d), whereas for a bit larger values of $\tilde{a}$ the surplus filters are filled with copies of (parts of) the motif (see figures 13e to 13g). Note that the motif which was also highlighted in the paper (figure 6d) appears in all results from figure 13b to 13g at least once. Only if $\tilde{a}$ is changed by more than one order of magnitude the results become significantly different and the motif is no longer detected (see figure 14). This indicates that it is sufficient to vary only the order of magnitude of $\tilde{a}$ in order to find a regime where motifs appear in the results and fine tuning $\tilde{a}$ is not necessary. This strategy is also the recommended strategy to find an appropriate sparsity parameter in SCC.

A similar behavior can be observed when $\beta_{\mathrm{KL}}$ is varied (see figure 15 for changes within an order of magnitude and figure 16 for larger changes). One can see similar effects as for the variation of $\tilde{a}$, but in the opposite direction: for smaller $\beta_{\mathrm{KL}}$ surplus filters are rather filled with copies of the motif whereas for larger values of $\beta_{\mathrm{KL}}$ the surplus filters are filled with background noise. This shows that it is usually sufficient to only tune one of the two - either $\tilde{a}$ or $\beta_{\mathrm{KL}}$ - in order to achieve good results.

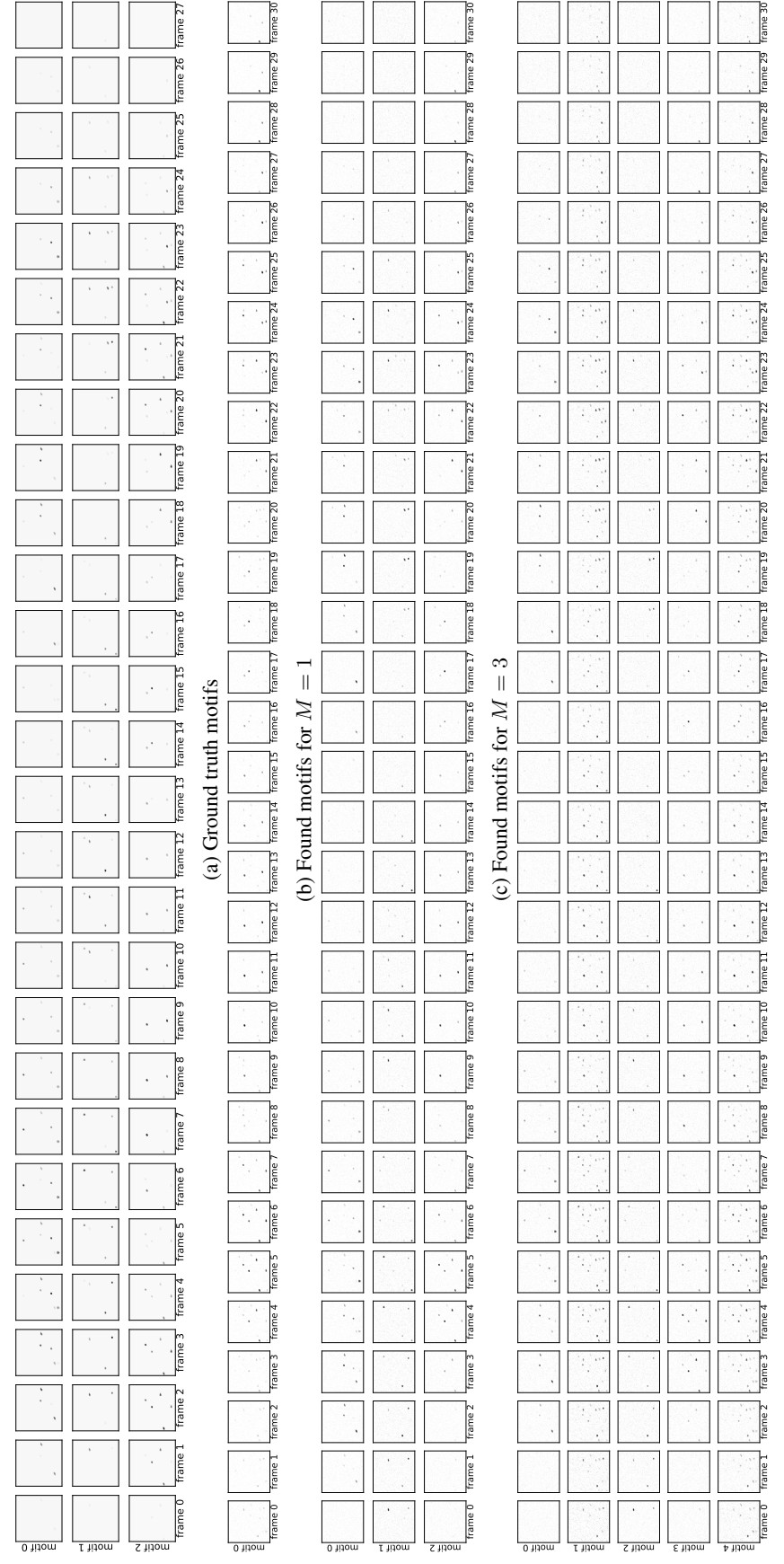

Figure 10: Results from the exemplary synthetic dataset discussed in the paper. (a) shows the three ground truth motifs. We also show the results of our analysis with fixed motif length ($F = 31$) for the different numbers of motifs $M = 1$ (b), $M = 3$ (c) and $M = 5$ (d).

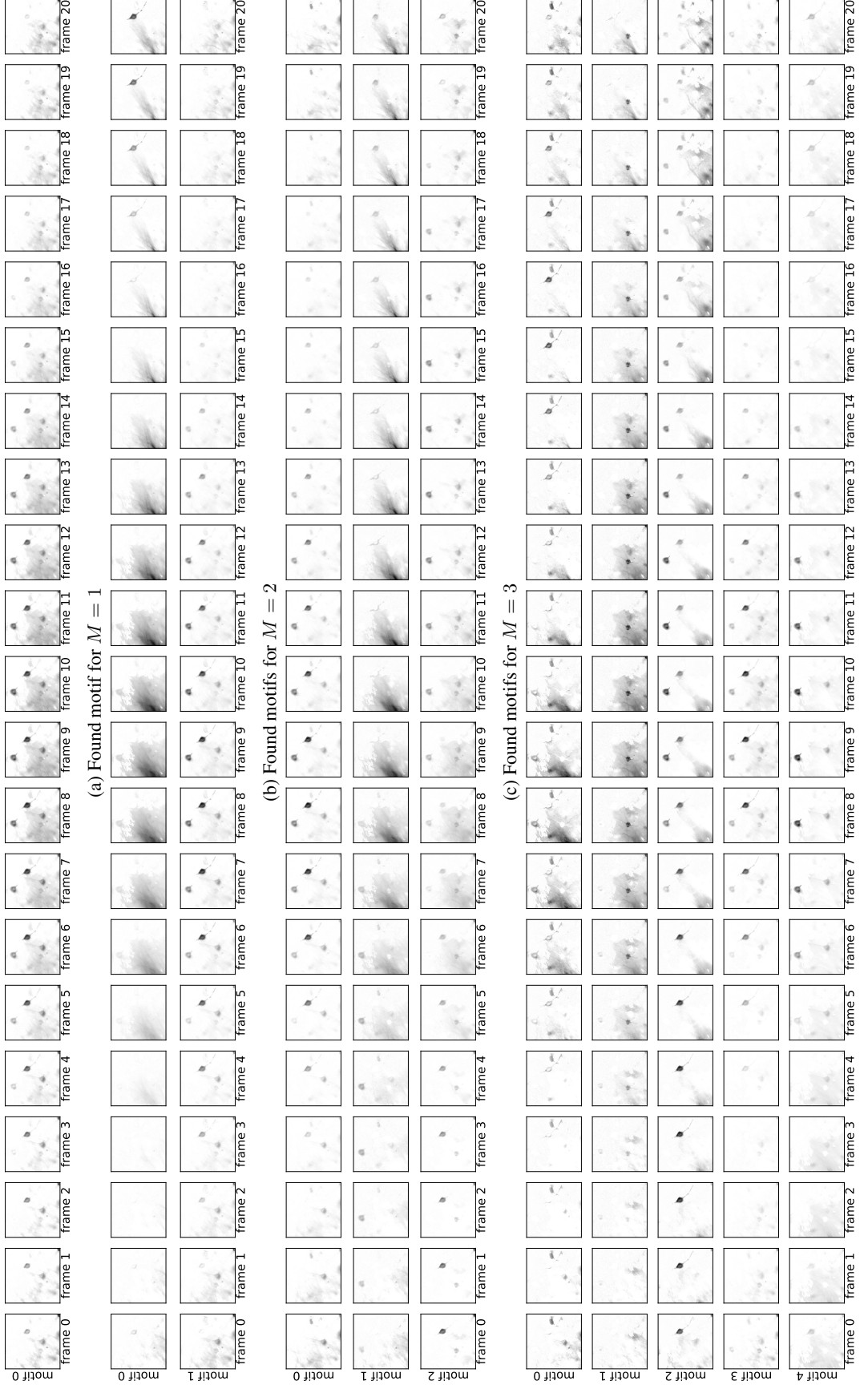

Figure 11: Results from dataset 1 with fixed motif length ($F = 21$) for the different numbers of motifs (a) $M = 1$, (b) $M = 2$, (c) $M = 3$, and (d) $M = 5$.

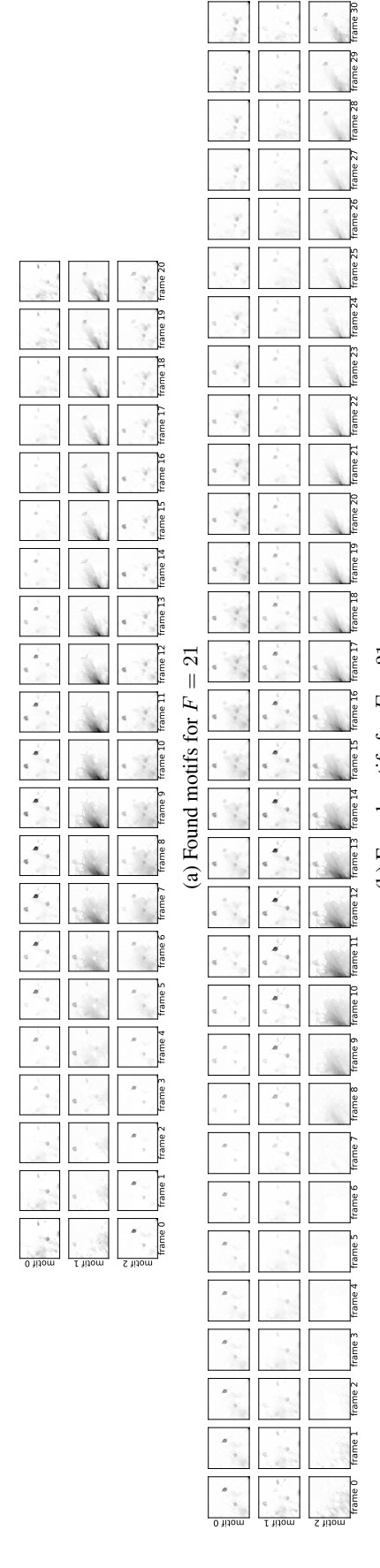

(a) Found motifs for $F = 21$

(b) Found motifs for $F = 31$

Figure 12: Results from dataset 1 with fixed number of motifs ($M = 3$) for the different motif lengths (a) $F = 21$ and (b) $F = 31$.

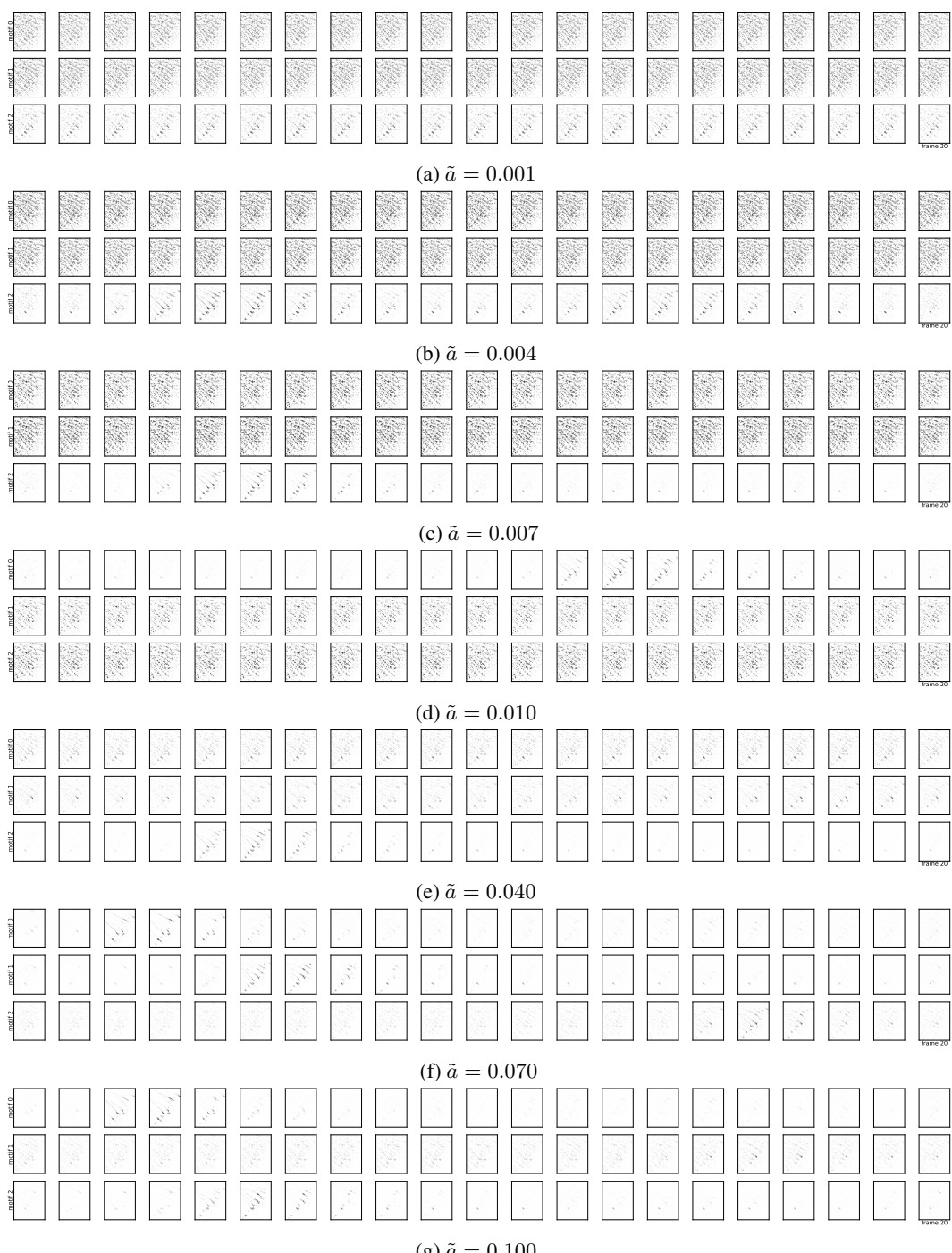

Figure 13: Motifs found on real dataset 2 for small changes of $\tilde{a}$. The parameter $\tilde{a}$ was increased in steps of $0.003$ from $\tilde{a} = 0.001$ (a) to $\tilde{a} = 0.010$ (d) and in steps of $0.030$ from $\tilde{a} = 0.010$ (d) to $\tilde{a} = 0.100$ (g).

## F MOTIF VIDEOS

In order to give the reader a better impression of what the used data and the motifs extracted as short video sequences would look like, we provide a few video files containing extracted motifs,

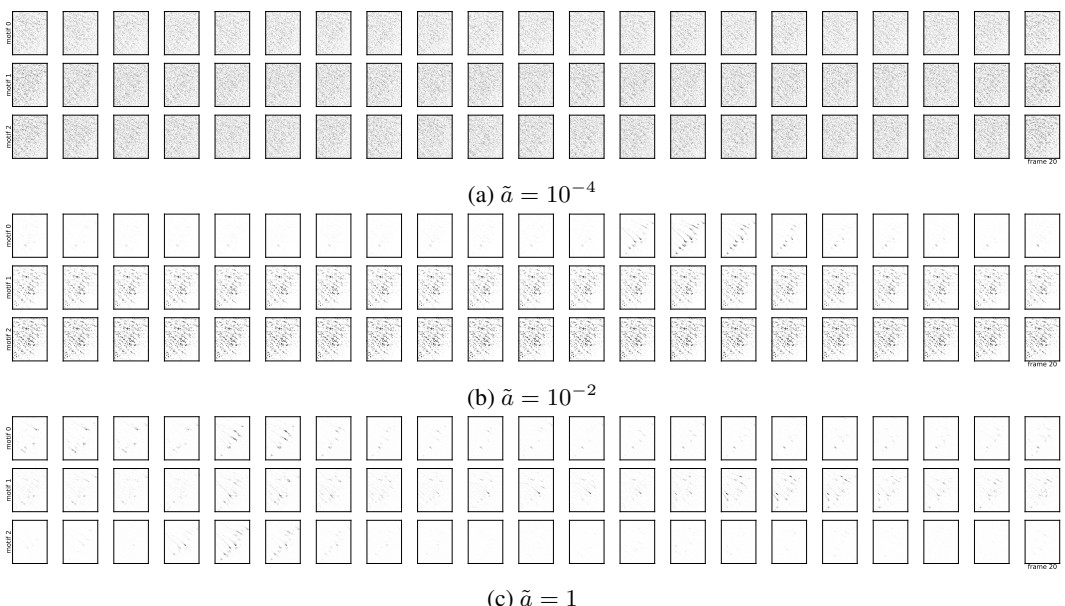

(a) $\tilde{a} = 10^{-4}$

(b) $\tilde{a} = 10^{-2}$

(c) $\tilde{a} = 1$

Figure 14: Motifs found on real dataset 2 for huge changes of $\tilde{a}$. The parameter $\tilde{a}$ was increased by two orders of magnitude in each step from $\tilde{a} = 10^{-4}$ (a) to $\tilde{a} = 1$ (c).

analyzed data and reconstructed videos at `https://drive.google.com/drive/folders/19F76JLn490RzZ4d7GxbWZoq6RdF2nt3w?usp=sharing`.

The reconstructed videos are gained by convolving the found motifs with the corresponding found activations. The videos are provided either in TIFF or MP4 format. Table 4 shows the names of the files together with short descriptions what each video shows. The videos corresponding to the synthetic dataset were generated with a frame rate of 30 fps and those corresponding to the real dataset with 10 fps.

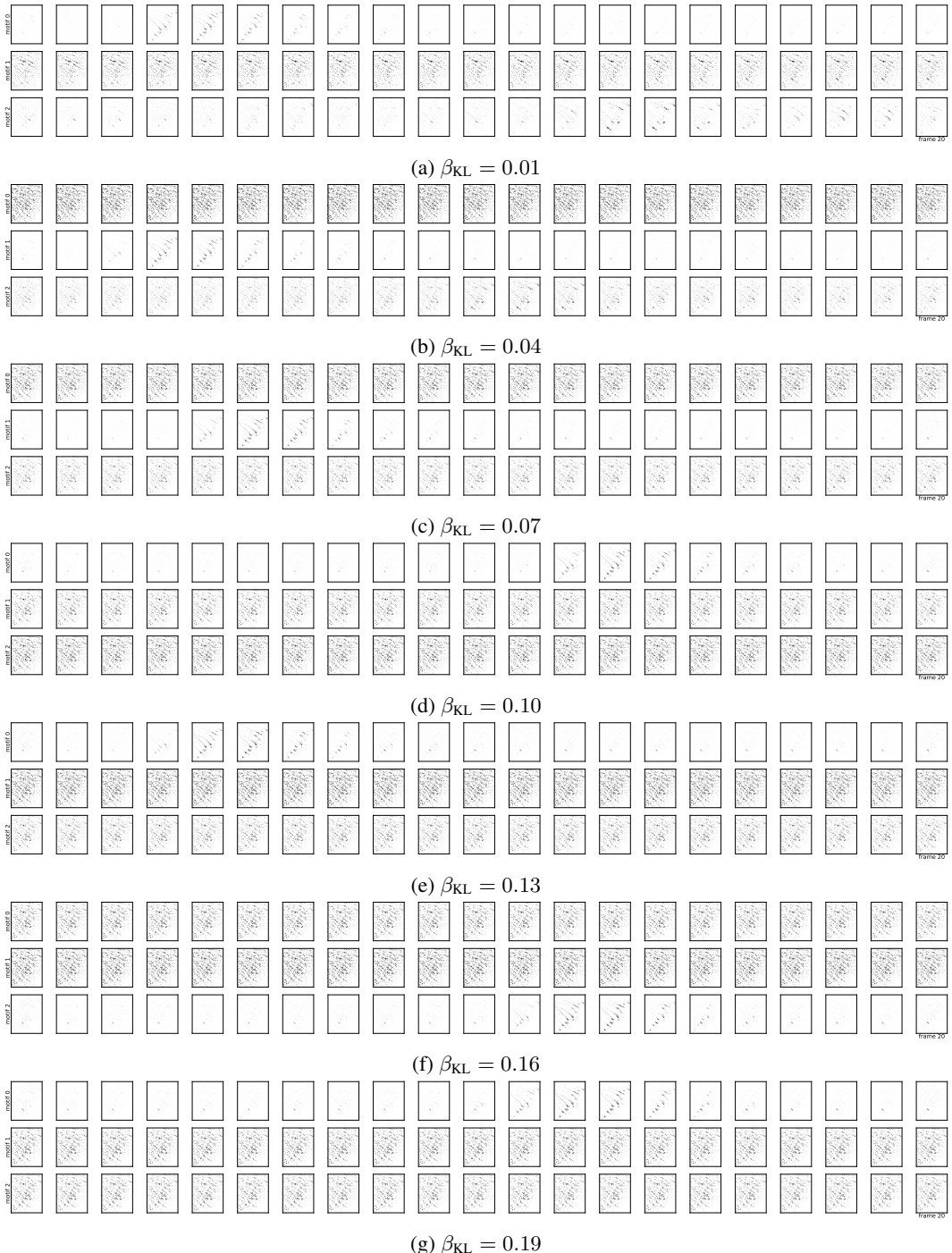

Figure 15: Motifs found on real dataset 2 for small changes of $\beta_{\text{KL}}$. The parameter $\beta_{\text{KL}}$ was increased in steps of 0.03 from $\beta_{\text{KL}} = 0.01$ (a) to $\beta_{\text{KL}} = 0.19$ (g).

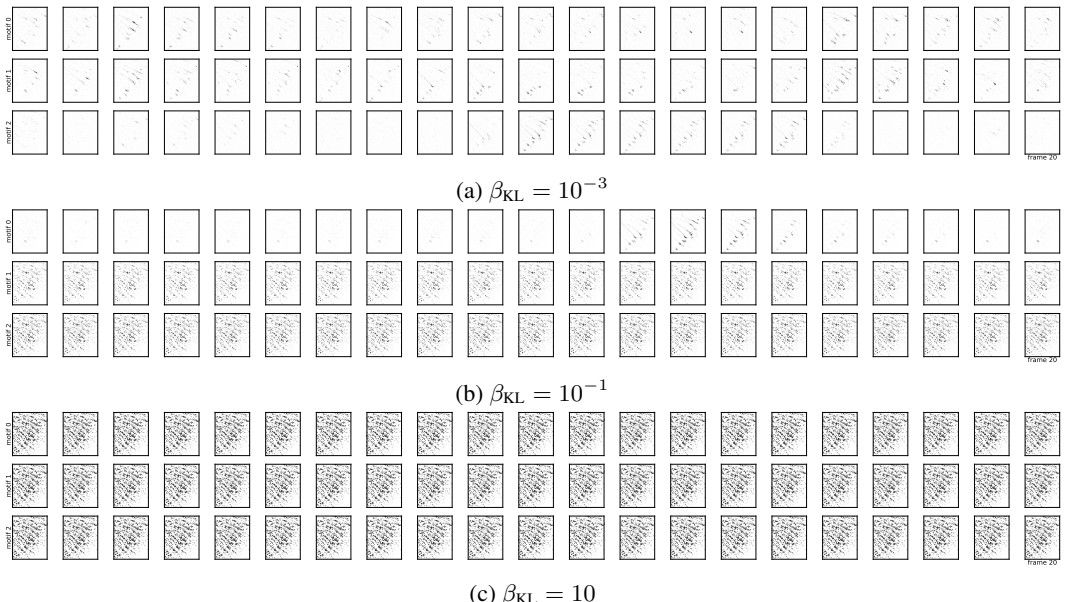

(a) $\beta_{\mathrm{KL}} = 10^{-3}$

(b) $\beta_{\mathrm{KL}} = 10^{-1}$

(c) $\beta_{\mathrm{KL}} = 10$

Figure 16: Motifs found on real dataset 2 for huge changes of $\beta_{\mathrm{KL}}$. The parameter $\beta_{\mathrm{KL}}$ was increased by two orders of magnitude in each step from $\beta_{\mathrm{KL}} = 10^{-3}$ (a) to $\beta_{\mathrm{KL}} = 10$ (c).

Table 4: Attached video files and descriptions. The used parameters for the analysis are the same as given in table 3 if not mentioned differently. The three different types of video are: *motif* showing a single motif; *parallel video* showing the original video from the dataset (upper left corner) and reconstructions from the found motifs; and *RGB video* showing a superposition of RGB values of the reconstructed videos from the three motifs found on the dataset. Additionally to the synthetic data example discussed in the paper (with 10% noise spikes), we also provide videos from a synthetic dataset with 50% spurious spikes.

| File name | dataset | video type | number of motifs $M$ | motif length $F$ |
|---|---|---|---|---|
| real_1_e1_l21_recon.mp4 | real dataset 1 | parallel video | 1 | 21 |
| real_1_e3_l21_motif_0.tiff | real dataset 1 | motif | 3 | 21 |
| real_1_e3_l21_motif_1.tiff | real dataset 1 | motif | 3 | 21 |
| real_1_e3_l21_motif_2.tiff | real dataset 1 | motif | 3 | 21 |
| real_1_e3_l21_recon.mp4 | real dataset 1 | parallel video | 3 | 21 |
| real_1_e3_l21_rgb.mp4 | real dataset 1 | RGB video | 3 | 21 |
| real_1_e3_l31_recon.mp4 | real dataset 1 | parallel video | 3 | 31 |
| real_1_e5_l21_recon.mp4 | real dataset 1 | parallel video | 5 | 21 |
| real_2_e3_l21_motif_0.tiff | real dataset 2 | motif | 3 | 21 |
| real_2_e3_l21_motif_1.tiff | real dataset 2 | motif | 3 | 21 |
| real_2_e3_l21_motif_2.tiff | real dataset 2 | motif | 3 | 21 |
| real_2_e3_l21_recon.mp4 | real dataset 2 | parallel video | 3 | 21 |
| real_2_e3_l21_rgb.mp4 | real dataset 2 | RGB video | 3 | 21 |
| synth_example_e3_l21_motif_0.tiff | synth. example | motif | 3 | 21 |
| synth_example_e3_l21_motif_1.tiff | synth. example | motif | 3 | 21 |
| synth_example_e3_l21_motif_2.tiff | synth. example | motif | 3 | 21 |
| synth_example_e3_l21_recon.mp4 | synth. example | parallel video | 3 | 21 |
| synth_example_e3_l21_rgb.mp4 | synth. example | RGB video | 3 | 21 |
| synth_50noise_e3_l21_recon.mp4 | synth. with 50% noise | parallel video | 3 | 21 |
| synth_50noise_e3_l21_rgb.mp4 | synth. with 50% noise | RGB video | 3 | 21 |

