# OpenReview forum: "LeMoNADe: Learned Motif and Neuronal Assembly Detection in calcium imaging videos"
_ICLR.cc/2019/Conference_

### Official Review · AnonReviewer3 · 2018-10-28
**i liked this paper last time i reviewed it, and i like it still :)**

**Rating:** 8
**Confidence:** 5

**Review:**

last time i had two comments:
1. the real data motifs did not look like what i'd expect motifs to look like. now that the authors have thresholded the real data motifs, they do look as i'd expect.
2. i'm not a fan of VAE, and believe that simpler optimization algorithms might be profitable.  i acknowledge that SCC requires additional steps; i am not comparing to SCC. rather, i'm saying given your generative model, there are many strategies one could employ to estimate the motifs.  i realize that VAE is all the rage, and is probably fine.  in my own experiments, simpler methods often work as well or better for these types of problems.  i therefore believe this would be an interesting avenue to explore in future work.

---

> ### Author Response · Authors · 2018-11-12
> **we liked your review last time, and we like it still ;)**
>
> We kindly thank the reviewer for his/her positive feedback. We would appreciate it if you could state again - as you did in your last review - why the problem we address is an important one and why quote "having an end-to-end procedure to learn motifs would be awesome". :)
> Of course there are different approaches in general to infer generative models. However, given our particular generative model our approach to do inference via VI in the form of a VAE is rather intuitive and stable compared to e.g. sampling based approaches which are computationally more expensive or EM based approaches which are usually less flexible. So we are not using VAEs just because they are all the rage right now (then we would use a GAN anyway ;) ). Nevertheless, we will of course continue doing research on this topic and will also investigate other approaches in future work.

---

> > ### Comment · AnonReviewer3 · 2018-11-13
> > **i like your response :)**
> >
> > to address why "end-to-end" is so important, the other strategy is both philosophically and statistically unpleasant.  philosophically, pipelining together many disparate processing stages creates all sorts of problems, as one cannot ever quite evaluate the impact of each decision on the final solution, because it is combinatorial. statistically, when one pipes together many algorithms, typically, the uncertainty associated with the output of each algorithm is lost when piping its MLE or MAP into the next algorithm.  so, if one were to report confidence intervals, they would be way too confident.  moreover, the goal in generating calcium imaging data is to discover motifs (or other interesting patterns), rather than, say, find all the spikes, which is really just a nuisance parameter.  so, piping things together results in optimizing each algorithm for the wrong metric.
> >
> > previous review is below:
> >
> > i am extremely knowledgeable in calcium imaging analysis, much less so in DL, VAE, etc. the problem that the authors address is very important and timely. the pre-processing of datasets for calcium imaging (and other modalities) is a mess, lots of steps, each with lots of parameters, inferences in downstream steps do not typically consider uncertainty in upstream tasks, etc. so, having and "end to end" procedure to learn motifs would be awesome.
> >
> > there are two aspects of this manuscript that i didn't love
> >
> > 1. is VAE really necessary here? it seems like an extension of the sparse dictionary learning stuff might be sufficient, and much simpler? i'm concerned because more complex methods have more algorithmic parameters to tune, and therefore, having one "end to end system" that is nearly as complicated as 3 disparate methods does not really one of the main motivating problems. the authors are clearly capable of doing a simpler thing. i realize this paper is not about that. but, for this paper, some discussion on why this approach was taken instead of the dictionary learning one, and some insight into the complications associated with actually running this method successfully on a new dataset would be highly desirable. fig 1 implies that lemonade is not just better, but also simpler, than other stuff. perhaps you could demonstrate or justify that a bit more?
> >
> > 2. the learned motifs in the simulated data looks like what i'd expect the motifs to look like. however, in the real data, they kind of look mostly like noise. i realize you did a bootstrap thing, etc. nonetheless, i am not convinced that you found legit motifs. specifically you even mention that the 3rd motif is not really even a motif at all, rather, just a single event. i wish there was a more convincing visualization or movie you could provide that made it really clear that you identified real biological motifs. i don't really have any good ideas for how to do that though.

---

### Official Review · AnonReviewer1 · 2018-11-02
**Interesting problem but the advantages of the model over other deep generative models are unclear**

**Rating:** 5
**Confidence:** 4

**Review:**

The paper proposes a VAE-style model for identifying motifs from calcium imaging videos. As opposed to standard VAE with Gaussian latent variables it relies on Bernouli variables and hence, requires Gumbel-softmax trick for inference. Compared to methods based on matrix factorization, the proposed method has the advantage of not requiring any preprocessing on the imaging videos. My main comments are as follows:

- How sensitive is the method to the choice of beta and other hyperparameters?  Compared to SCC which has fewer hyperparameters, how robust is the method?
- How does it perform on real data compared to methods based on spike time matrices? Do they generate similar motifs?
- The application of the method seems quite limited to calcium imaging videos and it does not provide comparison with other deep generative models for videos. Methods such as Johnson et al. NIPS 2016 (Composing graphical models with neural networks for structured representations and fast inference) can also be applied to calcium imaging datasets and can potentially infer the motifs.

I believe the problem of inferring the neural motifs is an interesting problem; however, I think this paper requires more work to it shows its advantages over other deep generative models for video data and also it’s performance on real data compared to SCC (or some other matrix factorization based approach).
-----------------------------------------------------------------------
The authors have addressed my comments about other deep generative models and hyperparameter sensitivity. However, I still think the paper is more suitable for other venues with readers from the neuroscience community. Hence, I change my rating to 5.

---

> ### Author Response · Authors · 2018-11-13
> **Response to AnonReviewer1**
>
> We appreciate the reviewers comments and will address them in the following and in the revised version of the manuscript which is already uploaded.
>
> Sensitivity to parameters:
> The main parameters that need to be chosen for each dataset individually are the maximum number of motifs and the maximum motif length. In appendix E.1 and E.2 we show the effects of over- and under-estimating these numbers for LeMoNADe and that they can be set to quite liberal values. Additionally, one of the sparsity parameters beta or â has to be adapted to the dataset. In appendix E.3 of the revised version we provide examples of different settings of â and beta, showing that they are complementary. This leaves us with three parameters that have to be adapted to a new dataset. For SCC also three parameters have to be chosen: number of motifs, motif length, penalty on l_1 norm of the assemblies = sparsity parameter.
> The examples in appendix E.3 also indicate that LeMoNADe's results are robust to small variations of â and beta and the results only change significantly when the parameters are varied by more than one order of magnitude. Peter et al. describe a similar sensitivity of SCC to the variation of their sparsity parameter.
> Other hyper parameters of LeMoNADe (e.g. temperatures of the BinConcrete relaxation, learning rate) do not need to be adapted to different datasets. We found that our default settings worked well for different kinds of data.
>
> Results on real data compared to SCC results:
> In appendix D.3 of the revised version we now show the results obtained with SCC on calcium traces of manually extracted ROIs from one of the datasets discussed in the paper. We also show, using traces extracted from the motif identified with LeMoNADe on the original dataset, that SCC and LeMoNADe find highly similar motifs on real data.
>
> Other generative models:
> As we mention in the related work section, a few deep generative models exist dealing with video data. However, to the best of our knowledge, none of these models is directly applicable to the task of detecting motifs with temporal structure in calcium imaging data.
> Indeed, Johnson et al. present an interesting generative model for the analysis of video data. However, we consider this model as not being able to identify motifs with temporal structure from calcium imaging data due to two limitations (for the detection of motifs in calcium videos) of the model by Johnson et al.:
> 1. Neuronal assemblies are expected to extend over multiple frames (depending on
> the frame rate of the recording this could be easily more than 20 frames). Since in Johnson
> et al.'s model the underlying latent process is a relatively simple first-order Markovian (switching) linear process, representing longer-term temporal dependencies will be very hard to achieve due to the usually exponential forgetting in such systems. In fact, Johnson et al.'s framework would need to be significantly extended, e.g. using LSTM units, to adapt their model for this task, which is a non-trivial task and could be considered to be a paper in its own right.
> 2. In the model of Johnson et al. each frame is generated from exactly one of K latent states. For calcium imaging, however, most frames are not generated by one of the motifs but from noise. While LeMoNADe has the chance to set the latent variables for noise frames simply to zero, Johnson et al.'s model would have to choose one motif as responsible for the frame even if it contains only noise. Moreover, LeMoNADe has the flexibility to also allow the different motifs to temporally overlap. This is also not possible in the model by Johnson et al., since they allow always only exactly one latent state for each frame.
> For this reason, we cannot compare to Johnson et al. on the task of detecting motifs in calcium imaging data. In the revised version of the manuscript we extended our citation of Johnson et al. with a short explanation why the model is not directly applicable to our setup of motif detection from calcium imaging data.
>
> The application is limited to calcium imaging data:
> The model and network architectures are indeed optimised for the task of detecting motifs in calcium imaging data. This is, however, no downside of the method. Calcium imaging is a method of first importance in neurophysiology. It allows the concurrent monitoring of the individual actions of thousands of neurons at the same time. As explained above, no other method is directly applicable to finding temporal motifs in calcium imaging data and in order to do so we had to adapt our method to the special properties of calcium imaging and neuronal assemblies. Nevertheless, our approach could also be adapted for detecting spatio-temporal motifs in data from other imaging techniques, such as voltage-sensitive dyes or functional magnetic resonance imaging (fMRI).

---

### Official Review · AnonReviewer4 · 2018-11-15
**Interesting ideas applied in the neural domain**

**Rating:** 8
**Confidence:** 4

**Review:**

Thank you for a pleasurable and informative read, I consider the writing and structure of the paper to be coherent and well written.

Given an end-to-end learning of neural motifs, a great deal of time can be avoided, reducing the several intermediary steps required to detect motifs from calcium imaging. This paper may very well improve researchers efficiency, in particular when working with calcium imaging. The question remain to what extent these ideas may be useful in other imaging modalities, i.e. fMRI.

My main critique would be to be more explicit about why the VAE you propose, is superior to other models in the generative modelling domain.

---

> ### Author Response · Authors · 2018-11-19
> **Response to AnonReviewer4**
>
> We are gratefull for AnonReviewer4's extra effort in order to provide us with a third review. We also thank him/her for the positive comments and for considering our paper to be well written and structured, and improving researchers efficiency.
>
> We agree that the application to fMRI and other imaging modalities makes for interesting future work.
>
> We took the reviewers comment into account and added a remark in the revised version at the beginning of section 3. The great benefit of this generative model in combination with the proposed VAE is the possibility to directly extract the temporal motifs and their activations and at the same time take into account the sparse nature of neuronal assemblies.

---

> > ### Comment · AnonReviewer4 · 2018-11-22
> > **Additional comments and questions**
> >
> > I've read the additional section, although I believe it would suffice, I would also add what inherent properties you chose to exploit from the VAE compared to alternatives such as GANs.  I would also like to add the following questions;
> >
> > - To what extent does your results depend on the adequate value of F (Your temporal frame dependency)? In your paper you experiment with different values of F, and for the synthetic data you assume up to 30 consecutive frames as the maximum length - Can you imagine a way of inferring F from the data rather than qualitatively assuming a fixed assembly firing structure?
> >
> > - How do you qualify the motifs found in the real data as a motif? In Section 4.2 you describe how you define F, looking for up to three motifs, and note that the model together with the SCC method suffer from potential false detections. In the same paragraph you note that this is qualitatively easily discarded by watching the videos. Are there heuristic references that may replicate your assessment of motifs when assessing alternative calcium imaging data? How did you chose the 70% thresholding value?
> >
> > My questions above highlight my need to understand the reproducibility of the results found in the paper, for which I am still, quite confident is of good quality.

---

> > > ### Author Response · Authors · 2018-11-23
> > > **Response to additional comments and questions**
> > >
> > > While GANs could be used to learn a generative distribution close to being indistinguishable from the observed data distribution, they are not directly applicable to the question we are targeting here. We are primarily interested in learning a certain constrained latent space structure for motif identification. In order to  get a posterior estimate for these latent variables, performing amortized inference via the proposed VAE is an intuitive and stable approach. This is not possible with a standard GAN.
> > >
> > > By fixing F to a certain value we do not assume a fixed assembly firing structure. The value F only provides an upper bound for the temporal extend of the firing pattern. Within the F frames we do not restrict the firing structure at all.
> > > An appropriate value of F mainly depends on the setup of the neurophysiological experiment and the research question in mind. For example, depending on the frame rate of the recording in one case it might make sense to look for motifs with temporal extend up to F=50 frames; while in another case with much lower frame rate motifs with temporal extend beyond 5 frames would be a big surprise. For this reason we intentionally left F as a parameter to be specified by the user. In cases where one is uncertain about the maximum temporal extent to be expected, there is no harm (other than computational effort) starting with a rather too large value of F.
> > >
> > > There are different potential outcomes of the method where the distinction between firing patterns that are actually repeating in the data and artefacts is easily possible with a background in neurophysiology (which we expect the users of our method to have).
> > > One example is shown in motif 1 found in real dataset 1. This motif shows extremely high luminosity in large parts of the imaging plane and individual cells can hardly be identified. When looking at the motif and also at the original video one easily sees that this comes from a single event at the beginning of the recording when the carbachol was washed in and the neuronal activity started.
> > > A second example are the motifs 1 and 2 found in dataset 2. When looking at them one sees that they also do not show individual cells but just randomly looking values. Therefore they can be easily identified as background noise.
> > > In addition to looking at the motifs themselves, one can also look at the activations. Even in the not-thresholded case motif 1 and 2 of dataset 1 show only one big peak in their activations. As the patterns we are looking for are defined by reoccurrence, this clearly identifies this “motifs” as artefacts. Thresholding the activations makes this finding even clearer.
> > > However, we prefer not to provide general instructions about when to discard a motif as this might also highly depend on the experimental setup and the scientific question.
> > >
> > > Neuronal assemblies are expected to show slightly variations in their firing and not every time the motif is active all cells might participate in the firing. Hence we expect the motif activation peaks to have different heights. Since the cells participating in any given motif will usually also fire outside the context of the motif, motif activation will also be non-zero elsewhere. We chose the threshold of 70% in order to show that for the motifs 0 found in the two datasets there are multiple time points in the recordings when a huge majority of the pattern is reactivated. Depending on the concrete scientific question it might be useful to apply a different threshold.

---

### Meta-Review · Area_Chair1 · 2018-12-14
**Good applied paper**

**Confidence:** 4
**Recommendation:** Accept (Poster)

**Metareview:**

This paper is about representation learning for calcium imaging and thus a bit different in scope that most ICLR submissions. But the paper is well-executed with good choices for the various parts of the model making it relevant for other similar domains.